# Oncolytic reprogramming of tumor microenvironment shapes CD4 T-cell memory via the IL6ra-Bcl6 axis for targeted control of glioblastoma

Jeffrey M. Grimes[1,2], Sadashib Ghosh [1], Shamza Manzoor[1], Li X. Li[1], Monica M. Moran[1,2], Jennifer C. Clements[1], Sherrie D. Alexander[1], James M. Markert [1,3] & Jianmei W. Leavenworth [1,3,4] ✉

Oncolytic viruses (OVs) emerge as a promising cancer immunotherapy. However, the temporal impact on tumor cells and the tumor microenvironment, and the nature of anti-tumor immunity post-therapy remain largely unclear. Here we report that CD4+ T cells are required for durable tumor control in syngeneic murine models of glioblastoma multiforme after treatment with an oncolytic herpes simplex virus (oHSV) engineered to express IL-12. The upregulated MHCII on residual tumor cells facilitates programmed polyfunctional CD4+ T cells for tumor control and for recall responses. Mechanistically, the proper ratio of Bcl-6 to T-bet in CD4+ T cells navigates their enhanced anti-tumor capacity, and a reciprocal IL6ra-Bcl-6 regulatory axis in a memory CD4+ T-cell subset, which requires MHCII signals from reprogrammed tumor cells, tumor-infiltrating and resident myeloid cells, is necessary for the prolonged response. These findings uncover an OV-induced tumor/myeloid-CD4+ T-cell partnership, leading to long-term anti-tumor immune memory, and improved OV therapeutic efficacy.

Malignant glioma (MG) remains an intractable problem with uniformly fatal outcomes in patients. It accounts for 30% of primary brain tumors in adults and an estimated incidence of ~25,000 new cases annually in the United States[1,2]. Glioblastoma multiforme (GBM) is the most commonly occurring MG in adults[1,3]. The course of this disease is marked by tumor recurrence with relentless regrowth, neurologic dysfunction, and, ultimately, death. The poor outcomes of GBM are attributed to its remarkable intratumoral heterogeneity, highly demanding metabolic requirements, and the specific environment of the central nervous system that is difficult to access by certain therapeutics. In addition, gliomas evade the immune system by excluding the immune infiltrates or competing with immune cells for nutrients

(e.g., cytokines or metabolites), creating a highly immunosuppressive "cold" tumor microenvironment (TME)[3,4]. With standard therapy (i.e., surgery, radiotherapy, and chemotherapy), the prognosis is still unfavorable with <5% of patients surviving beyond the 5-year mark after initial diagnosis[3,5,6]. Immune checkpoint inhibitors (ICI), such as anti-PD-1 and anti-CTLA-4, have shown promise in several types of cancer, but there is no obvious clinical benefit for GBM patients, largely due to the cold nature of TME[3]. Clearly, there is an unmet need to develop novel agents that can suppress GBM growth and recurrence.

To address this urgent medical need, we and others pioneered the concept of engineered oncolytic viruses for cancer. In particular, we focused on herpes simplex virus-1 (HSV-1) as an experimental virus for

[1]Department of Neurosurgery, University of Alabama at Birmingham, Birmingham, AL, USA. [2]Graduate Biomedical Sciences Program, University of Alabama at Birmingham, Birmingham, AL, USA. [3]The O'Neal Comprehensive Cancer Center, University of Alabama at Birmingham, Birmingham, AL, USA. [4]Department of Microbiology, University of Alabama at Birmingham, Birmingham, AL, USA. ✉e-mail: jleavenworth@uabmc.edu

treating brain tumors owing to its characteristics as a neurotropic, double-stranded DNA virus with a well-studied genome, allowing a rational approach to engineering[7,8]. The developed oncolytic HSV (oHSV) can selectively amplify in and kill tumor cells, including tumor stem cells, but not healthy cells, due to unique deletions in the $\gamma_1 34.5$ gene[7,9]. In addition, oHSV can boost anti-tumor immune responses, allowing the conversion of gliomas from "cold" to "hot" tumors that could become sensitive to other immunotherapies, such as ICI treatment[9–11]. To date, we have published the initial trials of "unarmed" G207 oHSV for MG, and also developed an "armed" oHSV-expressing human interleukin-12 (IL-12) (M032), which has been used in our phase I trial for recurrent MG patients (NCT02062827). Both G207 and M032 are safe and have shown promise with disease regression or stabilization in oHSV-treated recurrent MG patients, with some patients showing exceptional responses[12–15]. The oHSV-expressing murine IL-12 (M002) has also displayed a better therapeutic response than the parental control virus in preclinical GBM models[16,17]. However, the precise mechanisms for the enhanced anti-tumor immune responses to oHSV and the duration of this immunity remain largely unclear.

This study was initiated following the analysis of patient tumor specimens collected from our G207 phase 1b trial (NCT00028158)[15]. Although MGs typically lack effector T-cell infiltration, G207 markedly increases tumor-infiltrating T cells, which is associated with approximately half of patients demonstrating radiographic evidence of either disease stabilization or regression[18]. Moreover, RNA sequencing (RNA-seq) analysis of en bloc tumor resection has revealed that many T-cell-related genes are upregulated post-therapy and correlated with longer survival[15]. Our further analysis has pointed to the enrichment of MHCII-associated antigen-processing and presentation pathway post-therapy and in the longest survivor, which led us to hypothesize that anti-tumor CD4+ T cells are involved in response to oHSV therapy. Unlike CD8+ T cells, CD4+ T cells are conventionally recognized as T helper ($T_H$) cells and often undergo polarization into different $T_H$ subsets to accomplish tasks depending on the priming conditions and environment[19]. Their direct contribution to tumor control has also been recently appreciated[20,21]. Given the great degree of heterogeneity and plasticity, we investigate the nature of CD4+ T-cell response to GBM in response to M002 using syngeneic murine models of GBM. Our results support the requirement of CD4+ T cells for tumor control in an MHCII-dependent manner, and for sustained control of GBM, which depends on a memory CD4+ T-cell subset, whose signature is also correlated with better survival in our G207 clinal trial patients and TCGA-GBM patients.

## Results

### oHSV treatment reprograms the tumor microenvironment and increases MHCII on residual tumor cells

To begin analysis of the intratumoral immune changes induced by oHSV therapy, we re-evaluated the RNA-seq studies (GSE162643) of brain tumor biopsies from our phase 1b G207 clinical trial (NCT00028158)[15]. In addition to the significant changes of several immune-related genes and pathways induced by G207 therapy[15], we also found that genes in the antigen-processing and presentation pathway, including those encoding multiple MHCII molecules, were significantly upregulated post-therapy and positively correlated with survival (Supplementary Fig. 1a). To further understand the impact of oHSV therapy on the TME and which cell type(s) had upregulated MHCII, we decided to establish a syngeneic murine model of GBM using the GFP-expressing GBM cell line, GSC005, which displays stem-like features and generates tumors enriched with mesenchymal-like states[22], and closely recapitulates most of the immune-phenotypic signature of GBM patients compared to other cell lines[23]. B6 mice were implanted with GSC005 cells and then treated with M002 or saline vehicle control 11 days later. We then performed single-cell RNA-seq (scRNA-seq) analysis of the tumor and the TME at day 45 post-

treatment to facilitate our understanding of what cellular components post-oHSV-mediated tumor killing could contribute to the sustained beneficial outcome of this treatment. In a total of 15,241 cells after filtration and quality control, 17 clusters were identified and then grouped into 6 major cell types based on their representative marker genes[24,25] (Fig. 1a and Supplementary Fig. 1b–d). M002 treatment greatly changed the cellular composition of the TME with reduced tumor cells and microglia, but increased CD45+ cells, including T cells (Fig. 1b). Remarkably, M002 treatment upregulated several genes related to the MHCII pathway in a majority of intratumoral cell types, including the residual GBM tumor cells (specifically *H2-Ab1*) (Fig. 1c), which was likely attributed to the increased interferon α and γ response in these tumor cells (Supplementary Fig. 1e). Flow cytometry analysis of MHCII (I-Ab) expression further showed that although the proportions of microglia (CD45medCD11b+), tumor-infiltrating myeloid cells (CD45hiCD11b+), macrophages (CD45hiCD11b+Gr-1-F4/80+), dendritic cells (DCs) (CD45hiCD11c+MHCII+), CD45hiCD11c+ MHCII+CD11b+ DC subset and their MHCII-expressing populations were largely not significantly changed (except MHCII+ myeloid cells at day 21 post-treatment), the MHCII expression levels were significantly upregulated in myeloid cells, macrophages and DCs (Supplementary Fig. 2a–g). Interestingly, there were significantly more residual tumor cells (CD45-GFP+) expressing MHCII at higher levels, specifically on days 14 and 41 post-M002 treatment, than cells from mice treated with saline (Fig. 1d, e). We also noted that the proportion of MHCII+ tumor cells in saline-treated groups did not significantly change across each time point. In contrast, the frequency of MHCII+ tumor cells was increased compared to day 2 in the M002-treated group, particularly on days 14 and 41 (Fig. 1d, e). Overall, these results suggest that M002 treatment reprogrammed the TME and upregulated MHCII on tumor cells in vivo, potentially contributing to their increased immunogenicity.

### CD4+ T cells provide the M002 therapeutic benefit

The increased expression of MHCII on GBM tumor cells and intratumoral immune cells post-M002 treatment may suggest that CD4+ T cells play a role in mediating the M002 therapeutic efficacy. To test this proposition, we depleted CD4+ T cells in mice bearing GSC005 tumor prior to treatment with M002 or saline (Fig. 1f). Consistent with previous results[16], mice treated with M002 and an isotype control antibody for anti-CD4 significantly extended survival by twofold than mice treated with saline and the isotype control antibody ($P = 0.0009$, median survival 107 days vs 58 days) (Fig. 1f). In contrast, depleting CD4+ T cells abrogated the therapeutic benefit of M002 treatment, and the median survival was decreased to 50 days ($P = 0.0005$). The depletion of CD4+ T cells also significantly decreased the survival of saline-treated mice ($P = 0.002$) lowering their median survival to 45.5 days post-tumor implantation, which was significantly shorter than M002-treated mice with CD4+ T-cell depletion ($P = 0.006$). The CD4+ T-cell-mediated benefit was also observed in the M002-treated B6/GL261-PVRL1 model (Supplementary Fig. 2h), which was established from the parental GL261 cells by expressing the viral entry receptor, poliovirus receptor-related 1 (PVRL1, also known as nectin-1), to increase viral entry and replication (Supplementary Fig. 2i)[26], despite that GL261 cells are more immunogenic, and mice implanted with these cells have shorter median survival than mice implanted with GSC005 cells (-17 days vs -50 days)[27] (Fig. 1f). Interestingly, CD8+ T-cell depletion using the anti-CD8 depleting antibody 2.43 had no significant impact on the survival of saline-treated mice (Supplementary Fig. 2j). We also did not observe a significant effect of CD8+ T-cell depletion on the survival of M002-treated mice, although M002 treatment consistently prolonged mice survival compared to the saline-treated groups (Supplementary Fig. 2j).

M002 treatment significantly prolonged mice survival with some surviving over 60 days post-tumor implantation (Fig. 1f). We then evaluated if CD4+ T cells contributed to the protection against tumor

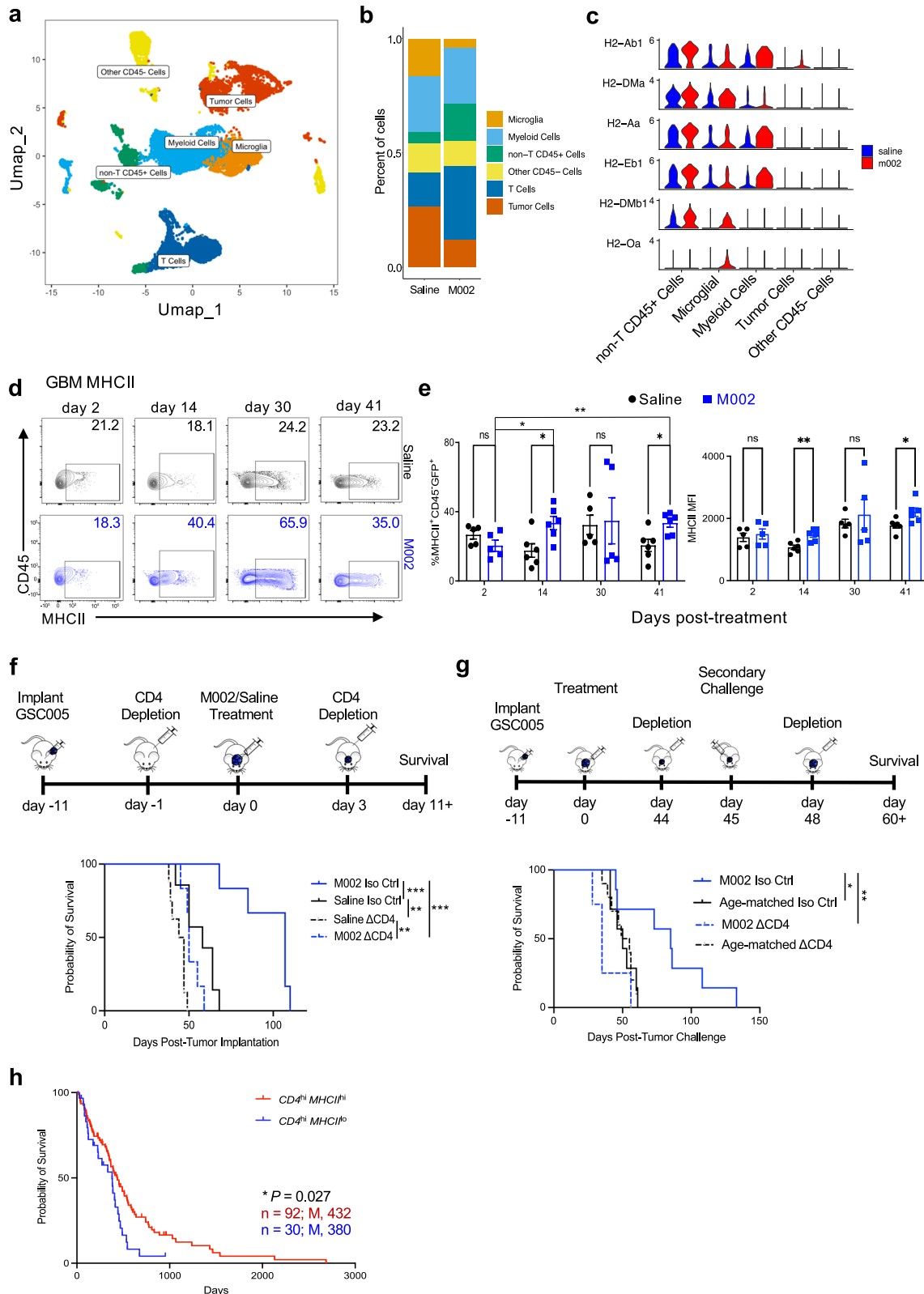

regrowth by re-challenging longer survivors in the M002-treated cohort at day 45 post-treatment through implanting GSC005 tumor cells in the contralateral hemisphere. Mice were treated with isotype control or anti-CD4 antibody to deplete CD4+ T cells before and after tumor re-challenge (Fig. 1g). Naive mice at an age that is matched to these M002-treated mice at the time of re-challenge were implanted with GSC005 tumor cells for the first time to serve as a control. M002-

treated mice that received the isotype control had a median survival of 85 days following the secondary challenge, which was a significant increase in survival when compared to the age-matched control mice ($P = 0.016$) that had a median survival of 50 days, and also had a significant increase in survival when compared to M002-treated mice with their CD4+ T cells depleted, which had a median survival of 35 days ($P = 0.0084$) (Fig. 1g). These results support that M002 treatment

**Fig. 1 | oHSV treatment increases MHCII on residual GBM tumor cells and CD4⁺ T cells provide therapeutic benefit. a–c** B6 mice were intracranially injected with GSC005 cells at day −11, and then injected with saline or M002 at day 0. CD45⁺ and CD45⁻ cells were sorted from tumors at day 45 post-treatment and subject to scRNA-seq analysis. **a** Sub-clustering of all cells presented by the UMAP plot. **b**, **c** Relative proportions of all clusters shown as stacked bar plots (**b**) and normalized expression of MHCII genes in all clusters shown as violin plots (**c**) across saline or M002-treated samples. T cells: cells with *Cd3e* and *Cd3g* > 1; Myeloid cells: all myeloid with DC excluded; non-T CD45⁺ cells: CD45⁺ cells with T, myeloid/microglia excluded; Other CD45⁻ cells: CD45⁻ with tumor cells excluded. **d**, **e** GSC005 tumor was established as in **a**, and brain tumor cells were analyzed at indicated days post-treatment by flow cytometry. Representative plots of MHCII expression in CD45⁻ GFP⁺ tumor cells (**d**), frequency of MHCII⁺CD45⁻ GFP⁺ cells (**e**, left), and the mean fluorescence intensity (MFI) of MHCII on CD45⁻ GFP⁺ tumor cells (**e**, right). Each dot represents an individual mouse in **e** on day 4.5 and day 30 ($n = 5$); on day 14 and day 41 ($n = 6$). **f** Schematic diagram of an experiment for CD4⁺ T-cell depletion at early time points. Mice were implanted with GSC005 cells at day −11, and treated with saline or M002 at day 0. At day −1 and day 3, mice received either anti-CD4 (ΔCD4, dotted lines) or an isotype control antibody (Iso Ctrl, solid

lines). Kaplan−Meier analysis of median survival as M002 Iso Ctrl ($n = 6$), 107 days; Saline Iso ctrl ($n = 7$), 58 days; M002 ΔCD4 ($n = 6$), 50 days and Saline ΔCD4 ($n = 8$), 45.5 days. **g** Schematic diagram of an experiment for CD4⁺ T-cell depletion at late time points following secondary tumor challenge. GSC005 implantation and treatment were performed as in **a**. At days 44 and 48, surviving M002-treated mice received either anti-CD4 (ΔCD4, dotted lines) or an isotype control antibody (Iso Ctrl, solid lines). At day 45, these mice were challenged with a secondary tumor in the contralateral hemisphere. Naive mice at an age that is matched to these M002-treated mice at the time of re-challenge were implanted with GSC005 tumor cells for the first time to serve as a control and were given either anti-CD4 or an isotype control antibody one day prior and three days after tumor implantation. Kaplan−Meier analysis of mice median survival as M002 Iso, 85 days ($n = 7$); age-matched Iso, 50 days ($n = 7$); M002 ΔCD4, 35 days ($n = 4$) and age-matched ΔCD4 ($n = 10$), 52 days. **h** Kaplan−Meier analysis of overall survival of patient cohorts expressing differential signatures of CD4 and MHCII based on their transcript levels from the TCGA-GBM dataset. n, patient numbers; M, median survival. Data represent one of two independent experiments (**e**–**g**). ns, no significance, *$P < 0.05$, **$P < 0.01$ and ***$P < 0.001$ (**e**, unpaired two-tailed Student's *t* test; **f**–**h**, log-rank test). Bars, mean ± SEM. Source data (**e**–**h**) are provided in the Source Data file.

induced a CD4⁺ T-cell-dependent control of tumor growth and regrowth, providing survival benefits in our GBM model.

Collectively, our preclinical data, consistent with the clinical results, pointed to a critical contribution of CD4⁺ T cells associated with the MHCII upregulation on tumor cells and intratumoral immune cells to GBM control and patient survival after oHSV therapy. Importantly, the beneficial contribution of the MHCII-CD4 axis to tumor control was also revealed by our analysis of TCGA-GBM datasets, which showed that GBM patients expressing higher *MHCII* (based on combined log-averaging of *HLA-DRA*, *HLA-DQA1,* and *HLA-DQB1* transcript levels) had better survival outcomes than their lower expression counterparts in the *CD4*ʰⁱ cohort (Fig. 1h).

## M002 treatment expands polyfunctional effector CD4⁺ T cells

We then explored how M002 therapy affected tumor-infiltrating lymphocytes (TILs), specifically CD4⁺ T cells, over the course of treatment using the B6/GSC005 and B6/GL261-PVRL1 glioma models, as described above. Profiling of T cells by flow cytometry revealed that mice treated with M002 had a significant increase in the total frequency and number of CD4⁺ T cells infiltrating both GSC005 and GL261-PVRL1 tumors at most time points post-M002 treatment compared to control mice (Fig. 2a and Supplementary Fig. 3a). The abundance of tumor-infiltrating CD4⁺ T cells was increased by 3–12-fold over the saline control in the GSC005 tumor and peaked around day 14 post-M002 treatment in both GBM models (Fig. 2a and Supplementary Fig. 3a). In general, CD8⁺ T cells were also increased in the GSC005 tumor after M002 treatment, albeit no significance was achieved (Supplementary Fig. 3b). Interestingly, almost no significant increases in natural killer (NK) cells were observed in both GBM models, and the frequency was even reduced at day 7 post-M002 treatment in the GL261-PVRL1 model (Supplementary Fig. 3c, d). These findings indicate a greater selective influence of M002 treatment on tumor-infiltrating CD4⁺ T cells than other lymphocytes typically correlated with a favorable prognosis in cancer.

As CD4⁺ T cells comprise both effector and suppressive regulatory T-cell (Treg) subsets, we analyzed these two compartments separately based on Foxp3 expression. A lower frequency of Foxp3⁺ Treg cells at all time points post-M002 treatment was observed compared to Treg cells in control mice (Fig. 2b). We did observe an increase in the total number of tumor-infiltrating Treg cells in M002-treated mice, but this was secondary to the overall increase of total TILs and CD4⁺ T cells (Fig. 2b). Again, the changes in Treg cells were consistent in both GBM models (Fig. 2b and Supplementary Fig. 3e). Analysis of Foxp3⁻ effector CD4⁺ T cells in both tumor models revealed that M002 treatment markedly increased the abundance of these cells expressing IFNγ and

TNFα with peaks at day 14 and day 7 post-treatment in GSC005 or GL261-PVRL1 tumors, respectively, and also increased cells expressing Granzyme B (GzmB) at early time points (Fig. 2c, d and Supplementary Fig. 3f). The increases in these inflammatory cytokines and cytotoxic molecules were consistent with the observations in longer survivals from our clinical trial[15]. In line with the augmented polyfunctional potential, intratumoral CD4⁺ T cells in M002-treated mice had a lower frequency of PD-1⁺Lag3⁺ population than CD4⁺ T cells in control mice (Fig. 2e), indicative of less exhausted status[28]. Taken together, these results demonstrate that M002 therapy expanded intratumoral CD4⁺ T cells with enhanced polyfunctional activity while diminishing the overall suppressive nature of the TME by reducing exhaustion and suppressive Treg cells.

To determine if M002-derived IL-12 contributed to the enhanced CD4⁺ T-cell response, we compared the CD4⁺ T-cell response in mice bearing GSC005 tumors that were treated with saline or M002 or R3659, the M002 noncytokine parent virus[17,29] at different time points post-treatment (days 4.5, 14 and 35). R3659 treatment increased the relative abundance of CD4⁺ T cells and decreased the relative abundance of Treg cells within the TME compared to saline-treated controls, albeit to a lesser extent than M002 treatment (Supplementary Fig. 4a, b). Importantly, the expression levels of GzmB and perforin were only significantly increased in CD4⁺ T cells from mice treated with M002, but not R3659, despite that both R3659 and M002 treatments had more CD4⁺ T cells expressing IFNγ and TNFα compared to saline controls at day 4.5 (Supplementary Fig. 4c–e). Both R3659 and M002 treatments also increased the proportion of CD8⁺ T cells expressing IFNγ and TNFα, albeit with no significance compared to the saline-treated group (Supplementary Fig. 4f). In addition, M002 but not R3659 treatment significantly increased the expression of GzmB, but not perforin, in CD8⁺ T cells at day 4.5 (Supplementary Fig. 4g, h). These results suggest that M002 treatment with the additional IL-12 had a better capacity to enhance the effector activity of CD4⁺ and CD8⁺ T cells (to a lesser extent) compared to its parent virus.

## Reprogrammed CD4⁺ T cells eliminate tumor cells in vitro in an MHCII-dependent manner

The antigen specificity of the expanded CD4⁺ T cells post-M002 treatment is unclear, as these cells could be reactive to HSV antigen or tumor antigen. To determine this, we leveraged the MHCII (I-Ab) tetramers based on the known tumor-associated antigen, tyrosinase-related protein-1 (Trp-1) and viral protein glycoprotein D (gD), expressed on GBM tumor cells and HSV-1, respectively[30,31]. We then established the B6/GL261-PVRL1 tumor model followed by treatment with saline, M002, or the control virus R3659, which generates

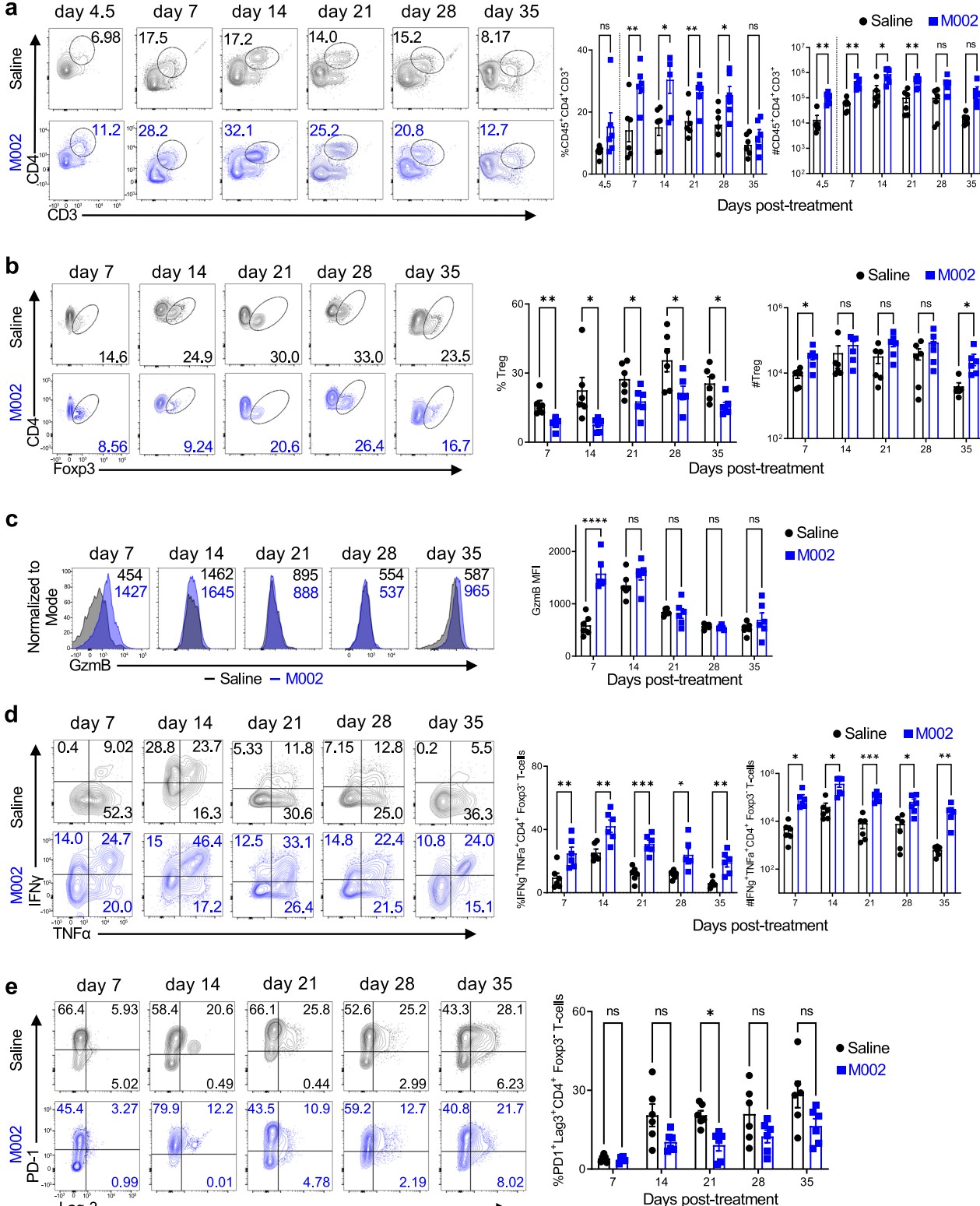

**Fig. 2 | M002 treatment expands polyfunctional effector CD4+ T cells.** Kinetic analysis of CD4+ T-cell abundance and function in brain tumors of mice implanted with GSC005 and treated with saline or M002 11 days later, as in Fig. 1a. **a, b, d** Representative plots (left), frequency (middle), and count (right) of CD45hiCD4+CD3+ T cells (**a**), CD45hiCD4+CD3+Foxp3+ Treg cells (**b**), and IFNγ+TNFα+CD45hiCD4+CD3+Foxp3− cells (**d**). **c** Representative histograms (left) and MFI (right) of GzmB expression in CD45hiCD4+CD3+Foxp3− cells. **e** Representative plots (left) and frequency (right) of PD −1+Lag-3+CD45hiCD4+CD3+Foxp3− cells. Each dot represents an individual mouse (n = 6 per group). Dotted vertical lines in **a** indicate that day 4.5 results were acquired on a flow cytometer different from the other time points. Data are pooled from two independent experiments. ns, no significance, *P < 0.05, **P < 0.01, ***P < 0.001, and ****P < 0.0001 (**a**–**e**, unpaired two-tailed Student's t test). Bars, mean ± SEM. Source data (**a**–**e**) are provided in the Source Data file.

reduced treatment efficacy compared to M002, as reported previously[16,17]. Using I-Ab/mouse Trp-1 113-126, I-Ab/HSV-1 gD 290-302, and the control tetramer I-Ab/human CLIP 87-101, we observed that M002 treatment at day 7 not only significantly increased intratumoral GzmB+CD4+ T cells, but also increased the proportion of these cells reactive to Trp-1 and gD, which was more than those CD4+ T-cell counterparts from mice treated with R3659 (Supplementary Fig. 5a, b). Almost no intratumoral GzmB+CD4+ T cells in saline-treated mice were positive for both tetramers (Supplementary Fig. 5a, b), suggesting that effector CD4+ T cells reactive to these antigens were specifically induced and expanded in response to oHSV treatment.

To directly demonstrate that intratumoral CD4+ T cells were reactive to tumor-associated antigen Trp-1, we isolated CD4+ T cells from tumors of GSC005-bearing mice that were treated with saline or M002 for 14 days, followed by co-culturing of these cells with mixed splenocytes containing cells pulsed with Trp-1 and those un-pulsed at a 1:1 ratio, distinguished based on the dose of CFSE labeling, with CFSElo as Trp-1-pulsed cells. There were fewer CFSElo cells in cultures with CD4+ T cells from M002-treated mice, indicative of more Trp-1-specific killing by these CD4+ T cells (Fig. 3a). Given the increased expression of GzmB, IFNγ and TNFα by M002-treated CD4+ T cells, and their tumor reactivity, we next determined if these CD4+ T cells were capable of mediating killing of tumor cells beyond their conventional helper activity. An in vitro killing assay was established using tdTomato+CD4+ T cells, characteristic of effector CD4+ T-cell population, isolated from GSC005 tumor of *GzmB*Cre*Rosa26*tdtomato mice post-treatment, as effector cells, and in vitro-cultured GSC005 tumor cells as target cells. The ratio of dead to living cells was used as a measure of cell-mediated killing. We observed that M002-treated CD4+ T cells were more capable of killing GSC005 tumor cells than CD4+ T cells from the saline control at all effector-to-target (E/T) ratios at both 14 and 35 days post-treatment (Fig. 3b, c). In contrast, M002-treated CD8+ T cells appeared to have reduced killing activity than saline-treated cells at higher E/T ratios (Fig. 3d), and CD4+ T cells had higher killing activity than CD8+ T cells at the E/T ratios higher than 5:1. Interestingly, the expression level of MHCII on GSC005 cells was significantly increased after co-culture with these ex vivo-isolated CD4+ T cells, and anti-IFNγ neutralization reduced this increase to some degrees, particularly in cells co-cultured with M002-treated CD4+ T cells (Supplementary Fig. 5c, d). Notably, the increase in M002-treated CD4+ T-cell-mediated killing was largely MHCII-dependent, as pre-incubation of tumor cells with an anti-MHCII substantially diminished the increases in their killing potential to the extent as those saline-treated CD4+ T cells, while MHCII blockade did not have a discernable impact on saline-treated CD4+ T cells (Fig. 3b). Consistently, neutralization of IFNγ, but not blockade of FasL or GzmB, diminished the M002-treated CD4+ T-cell-mediated killing (Supplementary Fig. 5e). Taken together, these results suggest that M002 treatment significantly expanded tumor-specific CD4+ T cells and enhanced the tumor-killing capacity of CD4+ T cells, but not CD8+ T cells, in our glioma models.

## Transfer of M002-programmed CD4+ T cells provides survival benefit

We next asked if M002-programmed CD4+ T cells could provide survival benefit in vivo. We then separately transferred an equivalent number (50,000) of CD4+ T cells isolated from M002-treated tumor or the saline-treated counterpart into *Tcra*−/− recipients that were then implanted with GSC005 cells at day 3 post-transfer (Supplementary Fig. 6a). As expected, *Tcra*−/− mice without T-cell transfer, as a control group, had the shortest survival, while transfer of CD4+ T cells from both M002- and saline-treated donor mice significantly extended survival (Fig. 3e), further supporting that CD4+ T cells were essential for GBM control and survival benefit. Most importantly, mice that received M002-treated CD4+ T cells had a median survival of 68 days compared to mice given saline-treated cells, which had a median

survival of only 54 Days, demonstrating a significantly prolonged survival (*P* = 0.001) (Fig. 3e). We also performed a second adoptive transfer experiment, in which CD4+ T cells were transferred into *Tcra*−/− mice harboring already-established tumors implanted 25 days prior (Supplementary Fig. 6a). Notably, mice transferred with M002-treated CD4+ T cells had a median survival of 27.5 days post-transfer, which was significantly longer than mice given saline-treated CD4+ T cells that had a median survival of 11.5 days post-transfer (Fig. 3f). These results again support our hypothesis that CD4+ T cells obtained from M002-treated mice harboring GBM demonstrated increased capacity to control established orthotopic GBM, and that the survival benefit was not solely attributable to the increased magnitude of the intratumoral immune response.

Leveraging the above first set of adoptive transfer system, we then asked whether transferred CD4+ T cells directly or indirectly controlled GBM, contributing to improved survival. We first profiled the innate immune cells that are implicated in the control of GBM growth, including microglia, NK cells, M1/M2 macrophages, DCs, and neutrophils, from both the tumor itself and spleens. The latter tissue served as a measure of the peripheral immune changes that could be influenced by transferred CD4+ T cells. At 10 days post-tumor implantation, we did not observe any significant difference in the total frequency of CD45medCD11b+ microglia, NKp46+NK1.1+CD3− NK cells, CD11b+F4/80Ly6C− MHCII+CD206− M1 macrophage, CD11b+F4/80+Ly6C− MHCII− CD206+ M2 macrophage, CD11b+F4/80− Ly6C+ monocytes, CD11c+MHCII+ DCs and DC subsets (CD103+ or CD11b+) as well as CD11b+Ly6G+ neutrophils in the tumor or spleen when comparing mice that received M002-treated CD4+ T cells to mice that received saline-treated cells (Fig. 3g and Supplementary Fig. 6b–f). There were also no significant differences for NK cells in the tumor and spleen in mice given CD4+ T cells from either saline- or M002-treated cohort compared to the control group without T-cell transfer (Supplementary Fig. 6b). However, intratumoral microglia, monocytes, and DCs were decreased, but neutrophils were increased in both CD4+ T-cell transfer groups compared to the no transfer control group. Interestingly, a similar trend for monocytes, DCs, and neutrophils in the spleen was noted (Fig. 3g and Supplementary Fig. 6c, d, f). Although the total macrophages in the tumor were similar in each group, and splenic macrophages were reduced in mice transferred with CD4+ T cells, a close analysis of intratumoral macrophage subsets revealed that the frequency of MHCII− CD206+ M2 macrophages was significantly increased in mice receiving effector CD4+ T cells compared to the no transfer controls, while the M1 subset did not differ substantially (Supplementary Fig. 6g). We also found increased intratumoral CD103+DCs and splenic CD11b+CD103−DCs in mice that received CD4+ T cells compared to the no transfer controls. In contrast, no changes for splenic CD103+DCs and intratumoral CD11b+CD103−DCs were observed among each group (Supplementary Fig. 6e). This profiling analysis suggests that programmed CD4+ T cells reshaped the TME and peripheral immune responses, although further analysis is needed to define the overall nature of these responses. However importantly, M002-treated CD4+ T cells influenced other immune cells in the tumor and periphery similar to saline-treated CD4+ T cells, which cannot account for the beneficial effects imparted by M002-treated CD4+ T cells.

We then assessed the direct effect of transferred CD4+ T cells on GBM tumor cells in vivo. We observed a significant reduction of CD45−GFP+ GSC005 tumor cells in mice given CD4+ T cells compared to the no transfer controls at day 10 post-tumor implantation, but a more significant reduction in tumor cell number in mice given M002-treated CD4+ T cells than mice transferred with saline-treated CD4+ T cells was noted (Fig. 3h). Moreover, the residual tumor cells from mice given M002-treated CD4+ T cells had significantly higher levels of MHCII than the other two groups (Fig. 3i). Unexpectedly, microglia in both mice transferred with CD4+ T cells had significantly lower MHCII expression

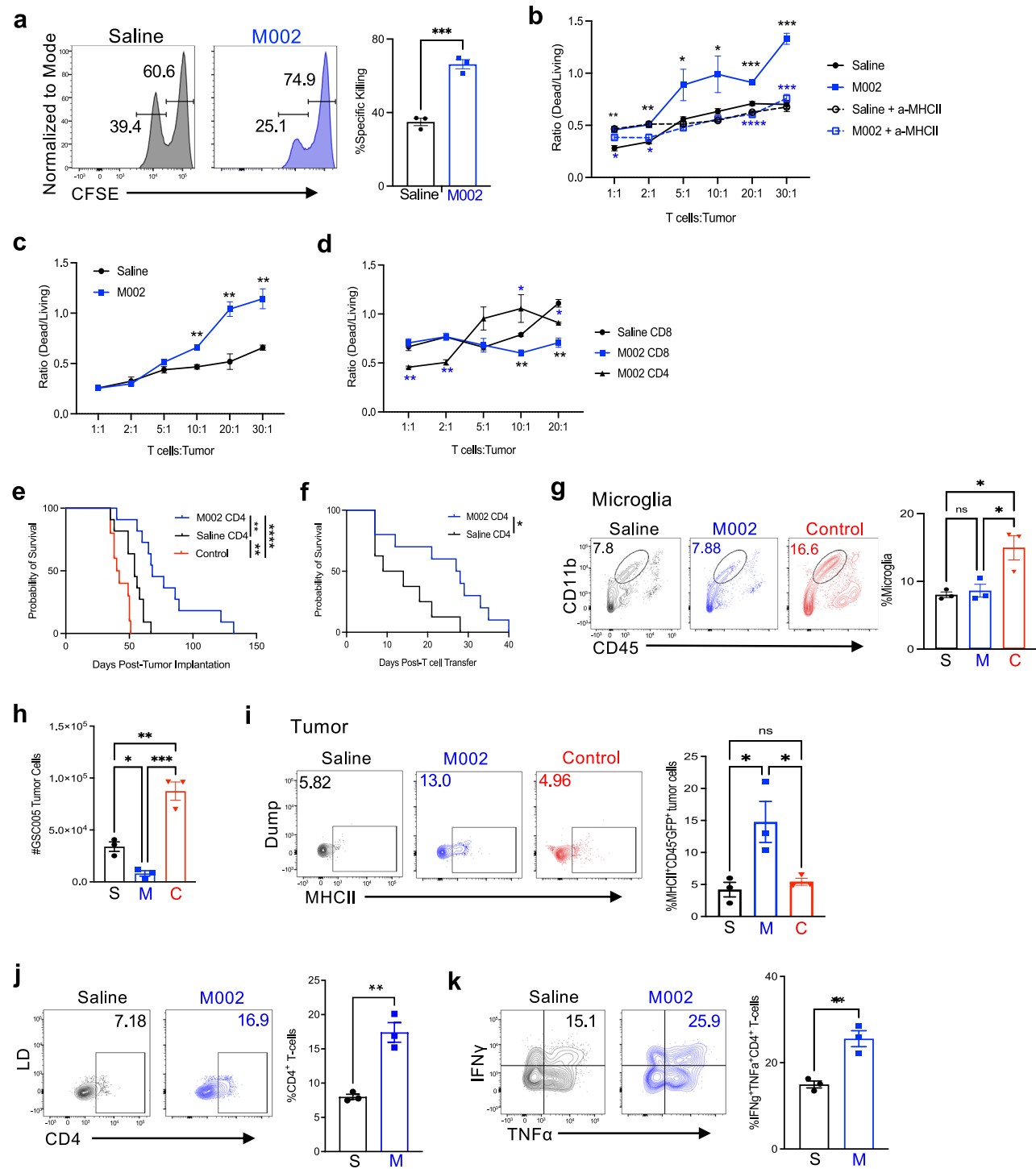

than microglia from the no transfer controls (Supplementary Fig. 6h). Finally, we evaluated CD4+ T cells themselves post-transfer. There were more CD4+ T cells found in the spleen and particularly in the brain tumor of mice transferred with M002-treated CD4+ T cells than in mice given saline-treated CD4+ T cells (Fig. 3j and Supplementary Fig. 6i). Most importantly, more of the former cells in the brain tumor expressed IFNγ and TNFα (Fig. 3k and Supplementary Fig. 6j). These results support that M002-programmed CD4+ T cells favored trafficking to the tumor and had enhanced expansion and functionality to control tumor growth, likely via an MHCII-mediated interaction with tumor cells, consistent with our earlier in vitro killing assays (Fig. 3b–d).

To further demonstrate the importance of tumor MHCII in mediating the anti-tumor activity of CD4+ T cells in vivo, we purified CD4+ T cells isolated from brain tumors of saline- or M002-treated mice and transferred them at the same number into MHCII KO (MHCII−/−) mice that were implanted with GSC005 tumor cells at 10 days earlier (Supplementary Fig. 6k). In this strategy, MHCII was only present on tumor cells. The anti-MHCII antibody was also injected into one group of mice given M002-treated CD4+ T cells to block the interaction between tumor MHCII and CD4+ T cells, based on the strategy reported previously[32]. We observed fewer surviving CD45−GFP+ tumor cells in mice transferred with M002-treated CD4+ T cells than mice given saline-treated CD4+ T cells (Supplementary Fig. 6l). The frequency of

**Fig. 3 | Reprogrammed CD4⁺ T cells eliminate tumor cells in an MHCII-dependent manner and provide survival benefit after transfer. a** GSC005 tumor was established and treated as in Fig. 1a. CD4⁺ T cells were isolated and enriched from brain tumors (pooled from 4 mice per group) at day 14 post-treatment, and then co-cultured with equal numbers of Trp−1 peptide-pulsed (labeled with a low concentration of CFSE, CFSE^lo) and un-pulsed (labeled with a high concentration of CFSE, CFSE^hi) splenocytes for 12 hr in triplicates for each group. Upper, representative histograms with percent cells in CFSE^lo or CFSE^hi gates. Bottom, quantification of percent Trp-1 peptide-specific killing. **b–d** GSC005 tumor was established and treated as in Fig. 1a. CD4⁺ T cells (**b, c**) or CD8⁺ T cells (**d**) were isolated and enriched from brain tumors (pooled from 4 mice per group) at day 14 (**b, d**) or day 35 (**c**) post-treatment, and then co-cultured with GSC005 tumor cells at an increasing ratio of T cells to tumor cells in triplicates for each group for 16 hr. Quantification of specific killing ratios is shown. In **b**, GSC005 tumor cells were pre-incubated with an anti-MHCII antibody (dotted lines, open symbols) or its isotype control antibody (solid lines, closed symbols) as controls for 30 min prior to co-culture with CD4⁺ T cells. **e–k** GSC005 tumor was established in *GzmB*^Cre*Rosa26*^tdTomato mice and treated as in Fig. 1a. CD45⁺ cells were isolated and enriched from brain tumors of saline- or M002-treated mice, and tdtomato⁺CD4⁺ T cells were sorted at day 14 post-treatment followed by transfer of 5 × 10⁴ cells into *Tcra*⁻/⁻ mice (**e**). 3 days later, *Tcra*⁻/⁻ mice were implanted with 5 × 10⁴ GSC005 cells (see Supplementary Fig. 6a). Kaplan–Meier analysis of median survival of *Tcra*⁻/⁻ mice receiving M002 CD4 (blue) (*n* = 11), 68 days; Saline CD4 (black)

(*n* = 11), 54 days and *Tcra*⁻/⁻ (control, without transfer of CD4⁺ T cells, red) (*n* = 10), 41 days. **f** GSC005 tumor cells were implanted into *GzmB*^Cre*Rosa26*^tdTomato and *Tcra*⁻/⁻ mice, separately. 11 days later, *GzmB*^Cre*Rosa26*^tdTomato mice were treated with saline or M002. 14 days post-treatment, CD45⁺ cells were isolated and enriched from brain tumors, and tdtomato⁺CD4⁺ T cells were sorted followed by transfer of 5 × 10⁴ cells into *Tcra*⁻/⁻ mice bearing already-established tumors for 25 days at the time of transfer (see Supplementary Fig. 6a). Kaplan–Meier analysis of median survival of *Tcra*⁻/⁻ mice receiving M002 CD4 (*n* = 10), 27.5 days and Saline CD4 (*n* = 8), 11.5 days. **g–k** Immune or tumor cells from brain tumors of *Tcra*⁻/⁻ mice (*n* = 3 per group) in **e** at day 10 post-tumor implantation. **g** Representative plots (left) and frequency (right) of CD45^medCD11b⁺ microglia. **h** Numbers of CD45⁻ GFP⁺ tumor cells. **i–k** Representative plots (left) and frequency (right) of MHCII⁺CD45⁻ GFP⁺ tumor cells (**i**), CD45^hiCD4⁺CD3⁺ T cells (**j**), and IFNγ⁺TNFα⁺CD45^hiCD4⁺CD3⁺ T cells (**k**). Each dot represents an individual mouse. Dump, dump gate. LD, live dead. S, transfer of saline CD4; M, transfer of M002 CD4; C, *Tcra*⁻/⁻ mice receiving no T-cell transfer as controls. All data represent one of two independent experiments. ns, no significance, **P* < 0.05, ***P* < 0.01, ****P* < 0.001, and *****P* < 0.0001 (**a–d; j, k** unpaired two-tailed Student's *t* test. **b–c**, black: M002 vs Saline; blue: M002 + a-MHCII vs M002. **d** black: M002 CD8 vs Saline CD8; blue: M002 CD4 vs M002 CD8. **e, f** log-rank test. **g–i** one-way ANOVA with Tukey's comparisons test). Bars, mean ± SEM. Source data (**a–k**) are provided in the Source Data file.

---

surviving tumor cells was increased in mice injected with the anti-MHCII antibody, which was not significant compared to mice given saline-treated CD4⁺ T cells (Supplementary Fig. 6l). These results suggest that the in vivo anti-tumor activity of CD4⁺ T cells required MHCII expression on tumor cells.

## The proper ratio of Bcl-6 to T-bet in CD4⁺ T cells navigates their enhanced anti-tumor capacity

CD4⁺ T cells are known to be heterogeneous, displaying different T_H features under certain conditions[19], which led us to define the nature of T_H subsets following oHSV treatment. We first analyzed the T_H1 subset, as increased IL-12 produced by M002 can preferentially induce T_H1 cell differentiation and T_H1 cells have a greater propensity to acquire effector function[20,21]. At the inception of the response (day 4.5), M002 treatment drove a much higher level of T-bet (MFI, 1752), the transcription factor for T_H1 subset, in tumor-infiltrating effector Foxp3⁻ CD4⁺ T cells, which was increased over 3-fold in comparison to saline treatment (MFI, 505) in mice bearing GSC005 tumors (Fig. 4a). However, the increase in T-bet expression was not sustained and no significant differences between these two groups were noted beyond day 7 post-treatment (Fig. 4a). Shortly after activation, CD4⁺ T cells undergo a binary fate choice between effector T_H1 and follicular helper T (T_FH) subsets, and increased levels of Bcl-6, a key transcription factor for T_FH, inhibit the acquisition of effector function of CD4⁺ T cells[33–35]. However, coinciding with the increased T-bet expression, Bcl-6 was also significantly upregulated in M002-treated CD4⁺ T cells at days 4.5 and 7 compared to saline-treated CD4⁺ T cells (Fig. 4b). No significant differences between these two groups were observed after 14 days of treatment (Fig. 4b). The above findings suggest that M002 treatment induced an early upregulation of T-bet in effector CD4⁺ T cells with the ratio of Bcl-6 to T-bet lower than that in the saline control (Fig. 4c). Despite that both T-bet and Bcl-6 levels were reduced later, the ratio of Bcl-6 to T-bet at day 35 post-M002 treatment was much higher than that in the saline control (Fig. 4c). Interestingly, the expression level of T-bet was only significantly increased in CD4⁺ T cells from mice treated with M002, but not R3659 (Supplementary Fig. 7a). Although both R3659 and M002 treatments increased Bcl-6 expression at day 4.5, the ratio of Bcl-6 to T-bet was not changed in R3659-treated cells compared to saline controls but a significant reduction was noted in M002-treated cells (Supplementary Fig. 7b, c), consistent with our above findings (Fig. 4c).

We then asked what is the contribution of the stage-specific expression of Bcl-6 relative to T-bet to the anti-tumor CD4⁺ T-cell response to M002 treatment. We first utilized mice with a T-cell-specific deletion of T-bet (*Tbx21*^fl/fl*CD4*^cre), attempting to increase the overall ratio of Bcl-6 to T-bet. The survival of these mice was compared to control mice (*Tbx21*^fl/fl) implanted with GSC005 tumor cells and treated with saline or M002 11 days later. Consistent with previous results (Fig. 1f), M002 treatment markedly extended mice survival compared to saline treatment with the median survival as 71 days versus 53 days (Fig. 4d). In contrast, deletion of T-bet in CD4⁺ T cells significantly reduced mice survival to 49 days and 43.5 days in M002- and saline-treated group, respectively (Fig. 4d and Supplementary Fig. 7d), indicating that the loss of T-bet in CD4⁺ T cells was deleterious to mice survival regardless of treatment (Fig. 4d). Of note, M002 treatment increased the frequency of CD4⁺ T cells infiltrating the tumor in both mouse strains compared to saline treatment (Supplementary Fig. 7e), which may partially explain why M002-treated *Tbx21*^fl/fl*CD4*^cre mice still had more prolonged survival than saline-treated *Tbx21*^fl/fl*CD4*^cre mice. In contrast, T-bet deletion reduced the abundance of CD4⁺ T cells compared to control *Tbx21*^fl/fl mice to a greater extent in the M002-treated cohort than in saline controls (Supplementary Fig. 7e). Analysis of effector activity of these tumor-infiltrating CD4⁺ T cells revealed that T-bet-deficient CD4⁺ T cells had markedly impaired functionality in both saline and M002 treatment groups, as the IFNγ⁺TNFα⁺ population and the expression of GzmB were significantly reduced (Fig. 4e, f). These results suggest that diminished T-bet expression, possibly the sustained increase of Bcl-6 to T-bet, could impair the survival benefit of M002 treatment.

We next decided to use mice with the tamoxifen-inducible CD4⁺ T-cell-specific deletion of Bcl-6 (*Bcl6*^fl/fl*CD4*^CreER) to define the impact of Bcl-6 deletion on the early and late stage of anti-tumor CD4⁺ T-cell response to M002 treatment (Fig. 4g). An early reduction of Bcl-6 along with the decrease in Bcl-6⁺CD4⁺ T cells prior to tumor implantation extended survival in *Bcl6*^fl/fl*CD4*^CreER mice compared to control *Bcl6*^fl/fl mice treated with M002, albeit no statistical significance. Additionally, no significant change was noted in the median survival of saline-treated control *Bcl6*^fl/fl mice when compared to saline-treated *Bcl6*^fl/fl*CD4*^CreER mice (Fig. 4g and Supplementary Fig. 7f, g). Consistently, M002 treatment significantly prolonged mice survival with increased intratumoral CD4⁺ T cells, IFNγ⁺TNFα⁺CD4⁺ T cells, and CD4⁺ T cells expressing GzmB and T-bet irrespective of Bcl-6 expression

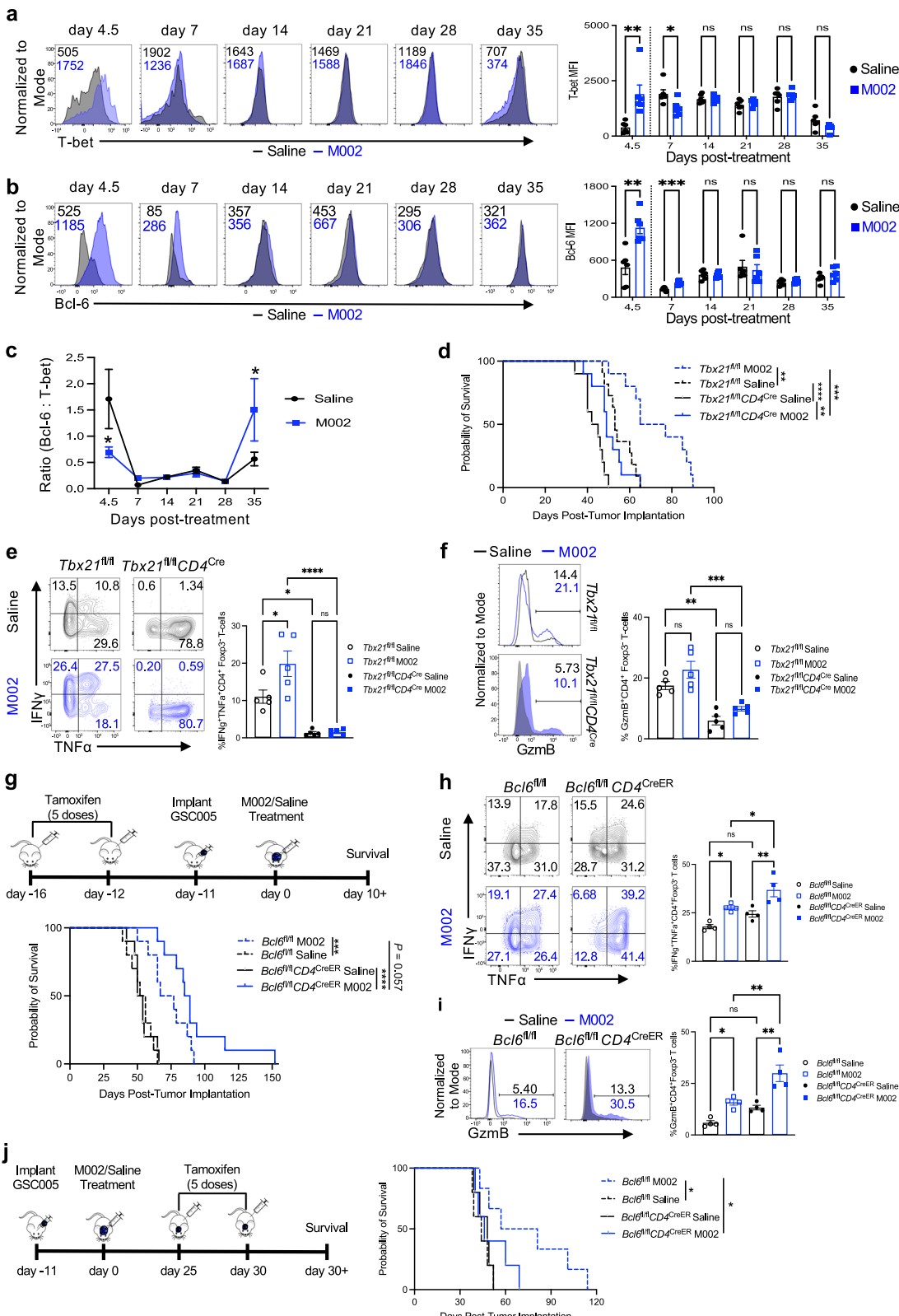

compared to saline treatment (Fig. 4g–i and Supplementary Fig. 7h, i). The proportions of IFNγ+TNFα+ and GzmB+ CD4+ T cells were also increased in cells expressing reduced Bcl-6 in M002-treated groups (Fig. 4h, i). These results suggest that an early decrease in Bcl-6 might contribute to the enhanced activation and effector function of CD4+ T cells after M002 treatment, but the survival benefit was suboptimal. Notably, induced Bcl-6 knockdown in CD4+ T cells at days 36-41 post-

tumor implantation, a later time point, had a negative impact on the survival of M002-treated mice with a median survival from 69 days in *Bcl6*fl/fl mice reduced to 46 days in *Bcl6*fl/fl*CD4*CreER mice, although this late depletion of Bcl-6 had no discernable impact on the survival of saline-treated cohorts (Fig. 4j). The late Bcl-6 knockdown also increased the expression of Tim-3 and T-bet (to lesser extents) in CD4+ T cells from M002-treated group, suggesting increased effector

**Fig. 4 | The proper ratio of Bcl-6 to T-bet in CD4$^+$ T cells navigates their enhanced anti-tumor capacity. a, b** Kinetic analysis of T-bet (**a**) or Bcl-6 (**b**) expression in CD4$^+$ T cells in brain tumors of mice ($n = 6$ per group) implanted with GSC005 and treated with saline (black) or M002 (blue), as in Fig. 1a. Representative histograms (left), and MFI (right) of T-bet (**a**) or Bcl-6 expression (**b**) in CD45$^{hi}$CD4$^+$CD3$^+$Foxp3$^-$ cells. Dotted vertical lines indicate that day 4.5 results were acquired on a flow cytometer different from the other time points. **c** The ratio of Bcl-6 to T-bet at indicated days as in **a, b. d–f** *Tbx21*$^{fl/fl}$ (dotted lines) or *Tbx21*$^{fl/fl}$ *CD4*$^{Cre}$ (solid lines) were implanted with GSC005 cells and treated with saline (black) or M002 (blue), as in Fig. 1a. Kaplan–Meier analysis (**d**) of median survival as *Tbx21*$^{fl/fl}$ M002 ($n = 10$), 71 days; *Tbx21*$^{fl/fl}$ Saline ($n = 11$), 53 days; *Tbx21*$^{fl/fl}$*CD4*$^{Cre}$ M002 ($n = 10$), 49 days and *Tbx21*$^{fl/fl}$*CD4*$^{Cre}$ Saline ($n = 10$), 43.5 days. **e, f** CD4$^+$ T cells from brain tumors from **d** were analyzed at day 17 post-treatment by flow cytometry ($n = 5$ per group). Representative plots (left) and frequency (right) of IFNγ$^+$TNFα$^+$ (**e**) or GzmB$^+$ (**f**) cells among CD45$^{hi}$CD4$^+$CD3$^+$Foxp3$^-$ cells. **g–i** *Bcl6*$^{fl/fl}$ (dotted lines) or *Bcl6*$^{fl/fl}$*CD4*$^{CreER}$ (solid lines) ($n = 10$ per group) were implanted with GSC005 cells and treated with saline (black) or M002 (blue), as in Fig. 1a. Tamoxifen was given prior to tumor implantation (**g**, upper). Kaplan–Meier analysis (**g**) of median survival as *Bcl6*$^{fl/fl}$ M002, 72 days; *Bcl6*$^{fl/fl}$ Saline, 54 days; *Bcl6*$^{fl/fl}$*CD4*$^{CreER}$ M002, 87 days and *Bcl6*$^{fl/fl}$*CD4*$^{CreER}$ Saline, 52 days. **h, i** CD4$^+$ T cells from brain tumors in **g** were analyzed at day 12 post-treatment by flow cytometry ($n = 4$ per group). Representative plots (left) and frequency (right) of IFNγ$^+$TNFα$^+$ (**h**) or GzmB$^+$ (**i**) cells among CD45$^{hi}$CD4$^+$CD3$^+$Foxp3$^-$ cells. **j** *Bcl6*$^{fl/fl}$ (dotted lines) or *Bcl6*$^{fl/fl}$*CD4*$^{CreER}$ (solid lines) were implanted with GSC005 tumor cells and treated with saline (black) or M002 (blue), as in Fig. 1a. Tamoxifen was given starting day 25 post-treatment (left). Kaplan–Meier analysis of median survival as *Bcl6*$^{fl/fl}$ M002 ($n = 6$), 69 days; *Bcl6*$^{fl/fl}$ Saline ($n = 5$), 44 days; *Bcl6*$^{fl/fl}$*CD4*$^{CreER}$ M002 ($n = 5$), 46 days and *Bcl6*$^{fl/fl}$*CD4*$^{CreER}$ Saline ($n = 5$), 48 days. All data represent one of two independent experiments. Each dot represents an individual mouse. ns, no significance, *$P < 0.05$, **$P < 0.01$, ***$P < 0.001$, and ****$P < 0.0001$ (**a, b** unpaired two-tailed Student's *t* test; **c** two-way ANOVA with Sidak's comparisons test; **d, g, j**, log-rank test; **e, f, h, i**, one-way ANOVA with Tukey's comparisons test). Bars, mean ± SEM. Source data (**a–j**) are provided in the Source Data file.

activity and possible exhaustion[36] (Supplementary Fig. 7j, k). Taken together, these results suggest that a proper expression of Bcl-6 relative to T-bet in CD4$^+$ T cells at early and late stages of anti-tumor responses was required for the sustained survival outcome of M002 treatment.

## The tumor-associated MHCII pathway facilitates the formation of a memory CD4$^+$ T-cell subset

The requirement of CD4$^+$ T cells for recall response (Fig. 1g) and the requirement of Bcl-6 expression in late-stage CD4$^+$ T cells for prolonged survival (Fig. 4j) prompted us to ask if M002 treatment had the potential to increase the memory feature of CD4$^+$ T cells in the tumor. Indeed, M002 treatment compared to saline controls increased CD103$^+$ or CD69$^+$CD103$^+$ CD4$^+$ T cells, populations with tissue-resident memory properties, in mice implanted with GSC005 or GL261-PVRL1, respectively (Supplementary Fig. 8a, b). To further dissect the nature of memory CD4$^+$ T cells induced by M002 treatment, we performed in-depth analysis of our scRNA-seq data of sorted CD45$^+$ TILs and CD45$^-$ cells from GSC005 tumors with a focus on intratumoral CD4$^+$ T cells at day 45 post-treatment with saline or M002. After quality control, data normalization, scaling, and dimensional reduction using the UMAP algorithm, 5 distinct clusters were revealed from a total of 1152 CD4$^+$ T cells with each annotated based on the published signatures for distinct T-cell states[37] (Fig. 5a and Supplementary Fig. 8c). A separate analysis of the M002 treatment group from the saline control group revealed the distinct distribution and representation of each cluster population, with a reduction in the fractions of cells in clusters 0 and 4 (*Foxp3* Treg cells) as well as cluster 1 (exhausted cells, *Pdcd1*, *Havcr2*, *Tox*), but an increase in the fractions of cells in clusters 3, an effector-like cluster based on its relatively high expression of *Tbx21* and *Cxcr6*, but relatively low levels of *Pdcd1* and *Havcr2* compared to cluster 1 (Fig. 5b, c and Supplementary Fig. 8c). Importantly, a substantial increase in the fraction of CD4$^+$ T cells in cluster 2 was noted (Fig. 5b, c). Comparing genes in cluster 2 to those in the other clusters showed that cells in cluster 2 expressed high levels of *Il6ra*, *Bcl6*, *Tcf7*, and *Bach2*, genes important for CD4$^+$ T-cell memory response[38–41] (Fig. 5d and Supplementary Data 1). GSEA analysis further supported the upregulation of pathways related to memory features, such as WNT and TGFβ signals, but downregulation of inflammatory pathways in cluster 2 (Fig. 5e and Supplementary Fig. 8d). Moreover, the overall memory signature in cluster 2 were higher in the M002-treated group than those in the saline controls (Supplementary Fig. 8e). Notably, *Bcl6* was largely represented in cluster 2, and *Bcl6*$^+$ cluster 2 cells were increased by M002 treatment (Fig. 5f and Supplementary Fig. 8c), suggesting that M002 treatment preferentially expanded a population of *Bcl6*$^+$CD4$^+$ T cells displaying the memory signature, while decreasing

suppressive and exhausted CD4$^+$ T-cell populations at the late time points.

Leveraging this scRNA-seq dataset, we further asked what types of cells could regulate the formation of such a memory CD4$^+$ T-cell population. CellChat analysis revealed that total CD4$^+$ T cells, particularly cluster 2 cells, interacted with microglia, myeloid, tumor cells, and other CD45$^-$ cells, with increased numbers and strength of interactions noted in the M002-treated group (Fig. 5g and Supplementary Fig. 8f). M002 treatment also increased multiple pathways and their related receptor/ligand pairs involved in these interactions, with MHCII/CD4 interactions consistently highly represented among all three interactions between total CD4$^+$ T cells, particularly cluster 2 cells, and microglia, myeloid or tumor cells, but not significantly upregulated between cluster 2 cells and non-T CD45$^+$ cells (e.g., DCs) (Fig. 5h, i and Supplementary Fig. 8g, h). Notably, only M002-treated CD4$^+$ T cells and cluster 2 cells, but not saline-treated counterparts, interacted with tumor cells via MHCII pathways, specifically *H2Ab1-Cd4* interactions (Fig. 5j and Supplementary Fig. 8i). Using NicheNet analysis, we predicted target genes in CD4$^+$ T cells that could be influenced by the interactions from microglia, myeloid and tumor cells (Supplementary Fig. 8j), and found that memory signature was highly enriched among these target genes in M002-treated CD4$^+$ T cells (Fig. 5k). These results suggest that increased interactions of CD4$^+$ T cells with microglia, myeloid and tumor cells may promote the acquisition of memory signature by CD4$^+$ T cells after M002 treatment, which could be facilitated by the MHCII signals provided from the tumor cells.

To further support the role of MHCII expression in shaping CD4$^+$ T-cell effector response and memory formation, we administered the anti-MHCII antibody into mice bearing GSC005 tumor with the treatment of saline or M002 to block the interactions between MHCII and CD4$^+$ T cells in vivo[32] (Supplementary Fig. 9a). Although MHCII blockade did not substantially change the overall abundance of CD4$^+$ T cells, there was a significant reduction of CD4$^+$ T cells expressing IFNγ and TNFα in mice treated with M002 and anti-MHCII compared to mice treated with M002 and its isotype control antibody (Supplementary Fig. 9b, c). MHCII blockade also reduced the Tcf-1 expression and the proportion of CD4$^+$ T cells expressing Bcl-6 and IL6ra in mice treated with M002 (Supplementary Fig. 9d, e). Interestingly, the above results were not obvious in saline-treated mice comparing those treated with anti-MHCII antibody to mice treated with its isotype control antibody. These findings support our in vitro findings (Fig. 3b) and CellChat analysis (Fig. 5g–k and Supplementary Fig. 8f–j), and provide additional support that the MHCII-mediated interaction was important for CD4$^+$ T-cell effector and memory response after M002 treatment in our experimental system.

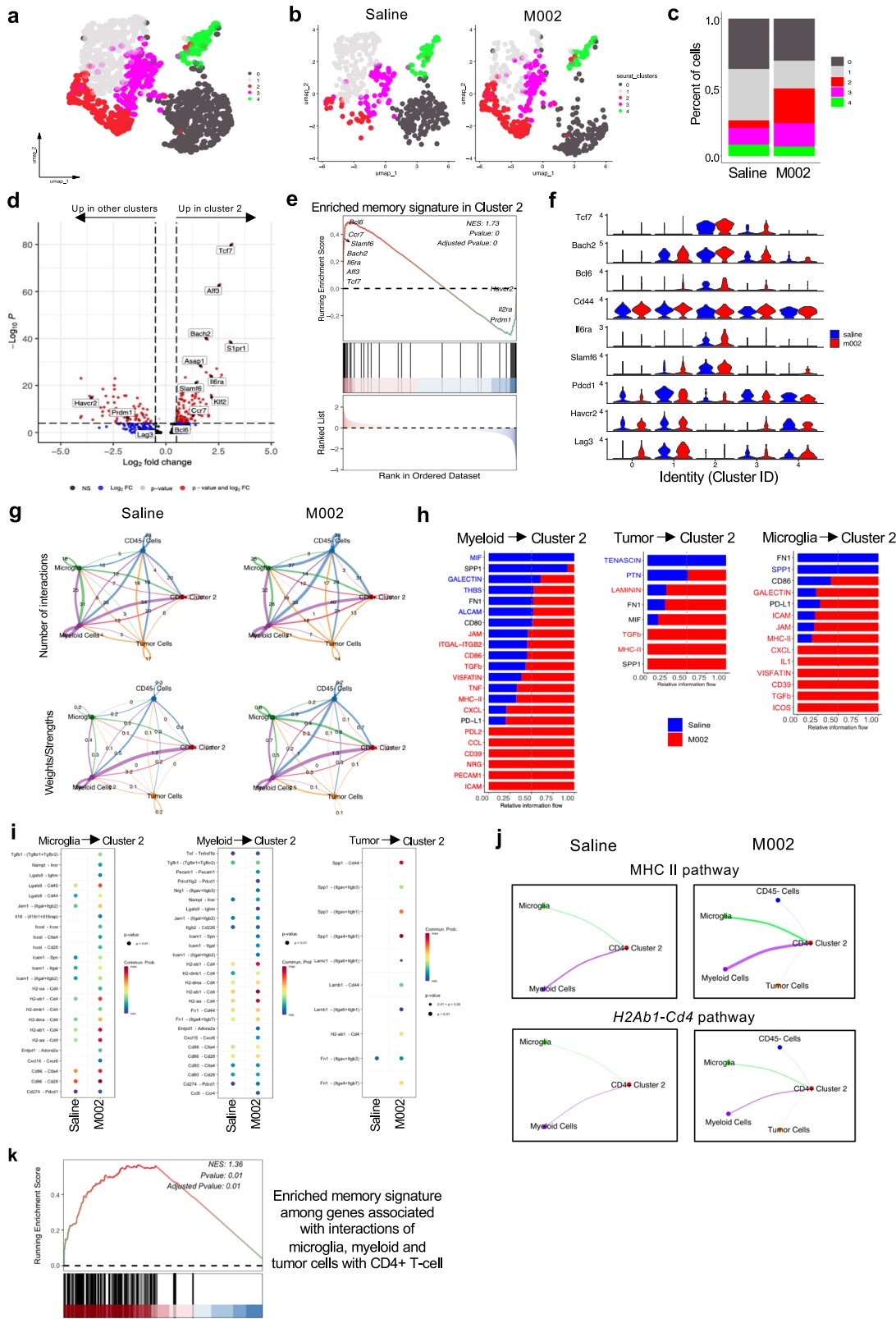

## The reciprocal IL6ra-Bcl-6 axis regulates the memory CD4+ T-cell differentiation

We next investigated the molecular mechanisms that could regulate the formation of the memory CD4+ T-cell population. The velocity mapping of *Foxp3*⁻CD4+ T cells showed that cells in the effector-like cluster 3 could differentiate into either cluster 1 or cluster 2 population (Fig. 6a). The corresponding two lineages were further revealed via

pseudotime trajectory analysis with lineage 2 giving rise to cluster 2 cells (Fig. 6b). Following the developmental path of lineage 2, genes related to effector activity and exhaustion, *Tbx21*, *Prdm1*, *Pdcd1*, and *Havcr2*, were downregulated, while genes associated with CD4+ T-cell memory responses, *Il6ra*, *Bcl6*, *Tcf7*, and *Bach2*, were upregulated, and M002-treated cells expressed the memory signatures at higher levels than saline-treated cells (Fig. 6c). Additionally, M002 treatment

**Fig. 5 | The tumor-associated MHCII pathway facilitates the formation of a memory CD4⁺ T-cell subset.** GSC005 tumor was established and treated, and scRNA-seq analysis of CD45⁺ cells was performed as in Fig. 1a. **a** Sub-clustering of CD4⁺ T cells. **b, c** Distribution of CD4⁺ T-cell clusters in the M002-treated group compared to the saline-treated group. **d** Volcano plots showing the differentially expressed (DE) genes in cluster 2 compared to the other 4 clusters with a two-sided non-parametric Wilcoxon Rank-Sum test with FDR adjustments for multiple comparisons. Red dots: genes with a fold change >2 and a *P* value < 0.05; blue dots: genes with a fold change >2 and a *P* value > 0.05; black and gray dots: non-significant genes based on fold change and *p* value cutoffs. Genes related to T-cell memory signature and ICI are denoted. **e** GSEA analysis for memory T-cell signature for cells in cluster 2 ordered by the ranked metric of all DE genes. **f** Expression of T-cell memory signature and ICI genes across the five clusters in the M002-treated group compared to the saline-treated group. **g–j** CellChat analysis of cell-cell interactions with CD4⁺ T-cell cluster 2 specifically. **g** Plots of cellular communication in indicated cell types based on the number or weights of ligand-receptor interactions (saline vs. M002). The width of the edge is proportional to a number of interactions or weights. **h** Horizontal stacked bar plots showing the relative information flow within inferred ranked pathways based on the differences of overall information flow within the indicated cell types (saline vs. M002). Red and blue denote pathways enriched in M002 and saline groups, respectively. **i** Bubble plots showing cell communication of indicated cell types based on the indicated ligand-receptor pairs enhanced in the M002-treated group as compared to the saline group using one-sided permutation testing under a null model corrected for multiple testing using FDR. The color scale denotes communication probability and bubble size corresponds to *P* value. **j** Plots of cellular communication of the MHCII pathway and *H2ab1-Cd4* interaction in saline and M002-treated groups. The width of the edge is proportional to the weight of interactions between indicated cell types. **k** The memory pathway was enriched from the GSEA analysis of predicted target genes in CD4⁺ T cells that could be influenced by the interactions from myeloid cells, microglia or tumor cells. **e, k** GSEA test statistic of enrichment score (ES, one-sided) was used to calculate the normalized enrichment score (NES) using a one-sided permutation test with FDR adjustments for multiple comparisons.

substantially changed the genes expressed along with the lineage 2 differentiation path compared to saline controls (Supplementary Fig. 10a, b and Supplementary Data 2). Notably, GSEA analysis of genes with their pseudotime values matching *Bcl6* upregulation following the lineage 2 trajectory revealed the significant enrichment of memory pathway in the M002-treated group (Fig. 6d and Supplementary Data 3). These results further support that M002 treatment promoted the formation of a *Bcl6*-associated memory subset.

The co-regulation of *Il6ra* and *Bcl6* following the memory cell differentiation post-M002 treatment (Fig. 6c and Supplementary Fig. 10b), and the reported T-cell-intrinsic role of IL6ra in promoting CD4⁺ T-cell memory[41] led us to define the potential contribution of IL6ra expression to the regulation of this *Bcl6*-associated memory subset. Interestingly, M002 treatment upregulated IL6ra in CD4⁺ T cells, while reduced Bcl6 expression in *Bcl6*^fl/fl^*CD4*^CreER^ mice downregulated IL6ra expression (Fig. 6e), suggesting a reciprocal regulation of Bcl-6 and IL6ra in CD4⁺ T cells. Moreover, the proportion of CD4⁺ T cells expressing Bcl-6 and IL6ra along with the expression of Tcf-1 on CD4⁺ T cells, but not CD8⁺ T cells, was significantly increased in mice treated with M002, but not in mice treated with R3659 at day 35 post-treatment (Supplementary Fig. 10c–e). When CD4⁺ T cells were isolated from the GSC005 brain tumor at day 12 post-treatment and subjected to co-culture with irradiated splenocytes plus Trp-1 peptide for 80 hr, we observed the increased CD4⁺ T cells, including IL6ra⁺Bcl-6⁺ and Tcf-1⁺Bcl6⁺ populations, and elevated expression of Bcl-6, Tcf-1 and GzmB in M002-treated cells compared to saline control cells, while these increases were all significantly reduced in the co-culture supplemented with anti-IL6ra blocking antibody (Fig. 6f–h and Supplementary Fig. 10f). These findings suggest that IL6ra expression and its associated signals positively regulated CD4⁺ T-cell recall responses and Bcl-6-associated memory signature. Notably, the IL-6-related signaling pathways along with MHCII-associated pathways were also highly enriched and upregulated in GBM patients post-G207 therapy and positively correlated with survival (Supplementary Fig. 10g and Supplementary Data 4), while the proportion of CD4⁺ T-cell memory population was significantly correlated with longer survival, as revealed via CIBERSORT analysis of our G207 clinical trial RNA-seq dataset (Fig. 6i and Supplementary Data 5). Finally, analysis of TCGA-GBM datasets showed that among patients expressing higher transcript levels of *CD4* and *BCL6*, those expressing higher levels of memory signature, including *IL6RA*, *TCF7*, and *BACH2*, compared to those expressing lower levels were correlated with better survival (Fig. 6j and Supplementary Data 1).

Taken together, our results suggest that M002 treatment reprogrammed the TME and expanded tumor-infiltrating CD4⁺ T cells with enhanced anti-tumor activity. The upregulation of MHCII on tumor cells facilitated tumor control by CD4⁺ T cells. The increased interactions between tumor/innate (microglia/myeloid) and CD4⁺ T cells also promoted the memory CD4⁺ T-cell differentiation that was modulated by the IL6ra-Bcl-6 axis and the proper Bcl-6 expression over the course of treatment, leading to long-term anti-tumor immunity and survival benefit (Fig. 6k).

## Discussion

Multiple preclinical and clinical studies using variants of oHSV to treat GBM have shown promise[12–15,42–45]. However, the nature of an effective and sustained anti-tumor immunity post-treatment is not entirely clear. Our present study highlighted an integral role of CD4⁺ T cells in the durable control of GBM and in mediating improved oHSV therapeutic efficacy.

Oncolytic viruses, including oHSV, are able to mediate direct tumor lysis and reshape the cold TME by enhancing anti-tumor immunity while diminishing immunosuppression[12–15,42–45]. Consistently, M002 treatment led to tumor killing and the enhanced overall magnitude of the anti-GBM immune response in the TME. However, the upregulation of MHCII on residual GBM tumor cells that promoted anti-tumor immunity is a new finding. The increase of MHCII expression was likely a result of the increased interferon responses in tumor cells, which was partially attributed to the increased IFNγ produced by M002-activated CD4⁺ T cells, according to our in vitro culture and in vivo adoptive transfer assays. The consequence of MHCII upregulation on tumor cells is two-fold, which not only benefited the CD4⁺ T-cell-mediated targeting of tumor cells, but also facilitated the programming of CD4⁺ T cells with enhanced polyfunctionality and memory potential. Increased interactions and signals from microglia and myeloid cells could promote CD4⁺ T-cell differentiation. However, the increased immunogenicity of tumor itself, particularly MHCII-mediated interactions from tumor cells post-M002 treatment, was also contributive, as reported for the ICI treatment of other types of cancer[20,46,47]. Notably, no appreciable *MHCII/CD4* interactions were predicted for saline control tumor cells and CD4⁺ T cells, despite that these tumor cells expressed MHCII (albeit at low levels). Possibly, the antigen presentation by MHCII on tumor cells and the T-cell receptor (TCR) repertoire of CD4⁺ T cells are determining factors for such interactions, as suggested from our tetramer analysis (although limited) that showed almost no tetramer staining for effector CD4⁺ T cells in the saline control group.

The differentiation of CD4⁺ T cells is modulated by TCR activation, specific cytokine and co-stimulatory signals as well as transcriptional factors. Our study has established the contribution of M002-derived IL-12 to the enhanced CD4⁺ T-cell effector activity and memory potential through a kinetic analysis compared to the M002 non-cytokine parent virus. Consistent with other reports in the context of chronic stimulation[39,40,48], the relative expression of Bcl-6 to T-bet was

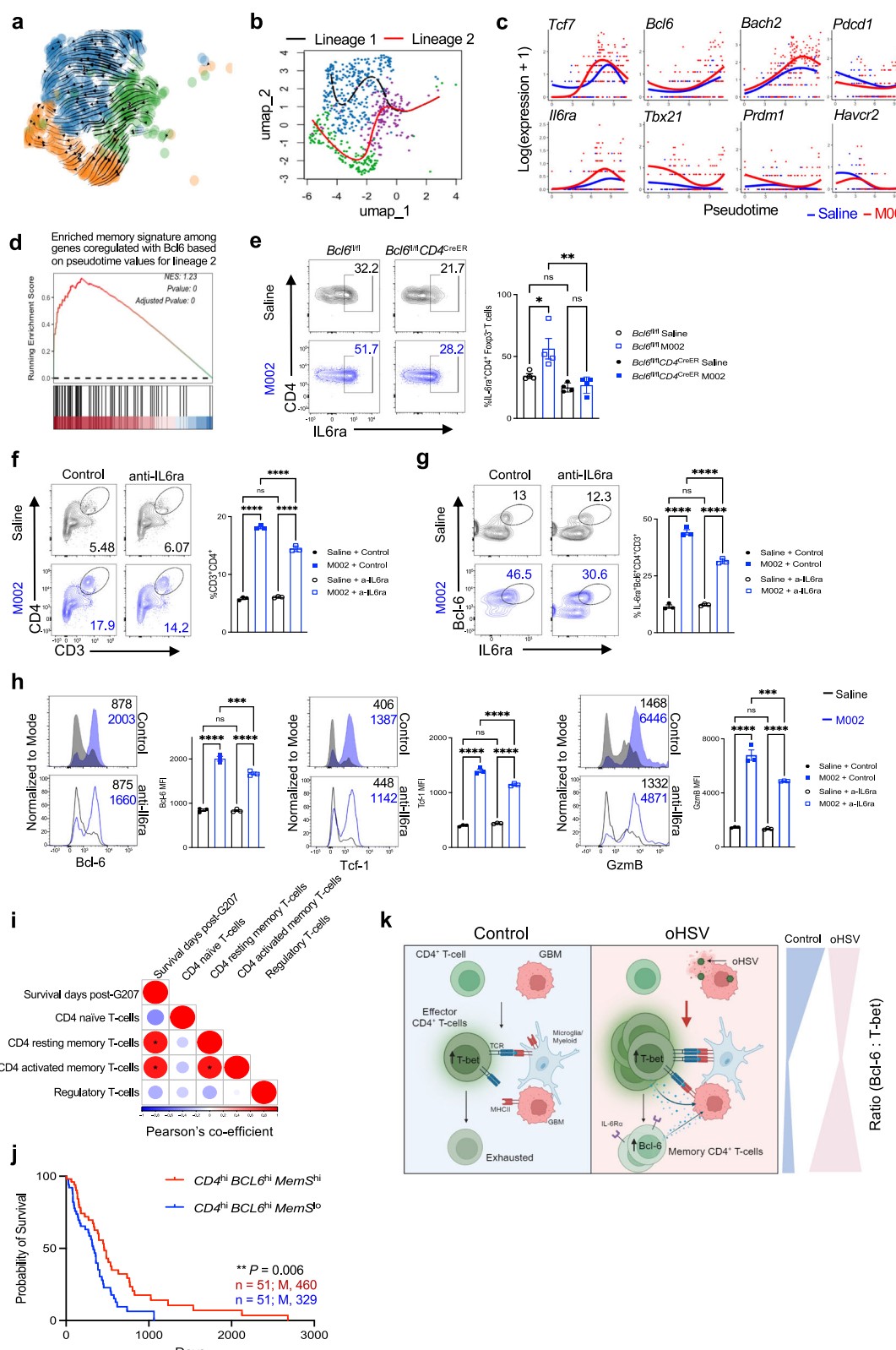

important for CD4⁺ T-cell fate decision over the course of M002 treatment. The inflammatory responses at the initial phase of M002 treatment allowed the concurrent upregulation of T-bet and Bcl-6. However, the ratio of Bcl-6 to T-bet had to be temporally optimized following the development of CD4⁺ T-cell response along with beneficial treatment outcomes. The sustained increase of Bcl-6 to T-bet secondary to the deletion of T-bet in CD4⁺ T cells was deleterious to

anti-tumor responses, and reduced Bcl-6 expression at the early stage of treatment only produced some survival benefit. Importantly, the upregulation of Bcl-6 to T-bet at the late stage, coinciding with the time when the anti-viral response waned but anti-tumor responses persisted, allowed for the expansion of a memory CD4⁺ T-cell subset that was responsible for long-lasting protective anti-tumor immunity. In contrast, the low expression levels of Bcl-6 to T-bet with sustained

**Fig. 6 | The reciprocal IL6ra-Bcl-6 axis regulates the memory CD4+ T-cell differentiation. a** UMAP plot shows velocity mapping of *Foxp3* CD4 clusters in Fig. 5a. Arrows indicate the direction of lineage differentiation for individual cells. **b** Slingshot lineages informed from initial velocity projection node in **a**. **c** The expression of genes related to memory and exhaustion following the lineage 2 trajectory (saline vs. M002) along pseudotime. **d** The memory pathway was enriched from the GSEA analysis of genes with their pseudotime values matching *Bcl6* upregulation following the lineage 2 trajectory. GSEA test statistic of ES (one-sided) was used to calculate the NES using a one-sided permutation test with FDR adjustments for multiple comparisons. **e** Flow cytometry plots (left) and frequency (right) of IL6ra+CD45hiCD4+CD3+Foxp3− T cells from brain tumors (*n* = 4 per group) in Fig. 4g. Each dot represents an individual mouse. **f**–**h** CD4+ T cells were isolated from the GSC005 brain tumor (pooled from 4 mice per group) at day 12 post-treatment, as in Fig. 1a, and subjected to co-culture with irradiated splenocytes plus Trp−1 peptide for 80 hr with anti-IL6ra or its isotype control antibody in triplicates for each group. Representative plots (left) and frequency (right) of CD45hiCD4+CD3+ T cells (**f**) and IL6ra+Bcl-6+CD4+CD3+Foxp3− T cells (**g**), and representative histograms (left) and MFI (right) of Bcl-6, Tcf−1 and GzmB expression in CD4+ T cells (**h**). Data represent one of two independent experiments (**e**–**h**). ns, no significance, *\*P* < 0.05, *\*\*P* < 0.01, *\*\*\*P* < 0.001, and *\*\*\*\*P* < 0.0001 (**e**–**h**, one-way ANOVA with Tukey's comparisons test). Bars, mean ± SEM. **i** Correlation of survival time post-G207 therapy with the presence of each CD4+ T-cell subset. *\*P* < 0.05 (Pearson's rho, two-sided correlation coefficient calculation without adjustment for multiple comparisons owing to only a few numeric variables). **j** Kaplan−Meier analysis of overall survival of patient cohorts expressing differential signatures of *CD4*, *BCL6*, and combined log-averaging of top 10 memory signature (*MemS*) genes (see Supplementary Data 1) based on their transcript levels from the TCGA-GBM dataset. *P* value is generated using log-rank test. n, patient numbers; M, median survival. ns, no significance. **k** Schematic presentation of identified mechanisms. Created in BioRender. Grimes, J. (2024) https://BioRender.com/z24v298. Blue shaded color indicates the cold, i.e., an immunologically cold tumor, and pink shaded color indicates hot, i.e., an immunologically hot tumor induced by oHSV treatment. Source data (**e**–**h**, **j**) are provided in the Source Data file.

effector activity may promote exhaustion and restrict memory formation, as reflected by the saline control group and the late Bcl-6 knockdown setting. The pattern of CD4+ T-cell response to M002 treatment resembled the scenario as reported previously, where only enough Bcl-6 is required for the generation of central memory CD4+ T cells in response to bacterial infection[48]. The expansion of this memory subset is likely contributed by the interplay of several factors, including the optimal TCR signals and interactions with other cells (e.g., tumor cells), as discussed above. Given the reported role of IL-6 and IL-6 signals in GBM progression[49] and in the regrowth of residual tumor after oHSV thearapy[50], it appeared contradictory to the finding that the increased IL6ra and IL-6 signals expanded this memory subset to mediate prolonged survival. However, the strength of IL-6 signals in CD4+ T cells versus tumor cells could differ, and targeting IL-6 to inhibit GBM tumor growth may need to consider its impact on CD4+ T cells, which needs to be further explored. Finally, the signature of Bcl-6-dependent memory CD4+ T cells overlapped with those CD4+ stem-like or progenitor cells or progenitor-exhausted CD8+ T cells in chronic infection or transplantation settings[39,40]. M002-programmed effector CD4+ T cells were capable of expanding and displaying effector activity after transfer or in vitro re-stimulation. However, further study is required to determine the fate of this Bcl6+ memory CD4+ T-cell subset.

The direct contribution of CD4+ T cells to tumor killing has recently attracted more attention[20,21]. In our models, M002 treatment programmed CD4+ T cells into polyfunctional cells with the capacity to control tumors, as evidenced by both in vitro killing assays and in vivo adoptive transfer approach. Moreover, results obtained from the CD4+ T-cell depletion assays supported the beneficial role of CD4+ T cells in the control of GBM growth and regrowth during M002 treatment. The tumor killing by M002-treated CD4+ T cells required MHCII signals, and also CD4+ T-cell-derived IFNγ that increased MHCII expression on tumor cells, forming a positive feedback loop. As transfer of M002-treated CD4+ T cells compared to transfer of a same number of saline-treated control cells preferentially reduced tumor cells, but not other immune cells, the MHCII-dependent anti-tumor activity may suggest an expansion of tumor-specific CD4+ T cells by M002 treatment, supporting the intrinsic epitope spreading induced by OV therapy[51,52]. Our results also suggest that the CD4+ T-cell-mediated anti-tumor response may not substantially require FasL and GzmB expression. However, the effector activity of CD4+ T cells may be subject to its differentiation stages in response to M002 treatment and the tumor microenvironmental cues, which requires future exploration, given that the increased GzmB (also perforin) expression was observed early after M002 treatment and upon in vitro and in vivo recall responses. In addition to the potential tumor killing by CD4+ T cells, our study does not rule out the possible contribution of other mechanisms to the

M002-mediated anti-GBM effects. CD4+ T-cell-derived IFNγ could increase microglial MHCII expression, as observed in our analysis, and then promote microglial phagocytosis of GBM tumor, in a way resembling the report of anti-CTLA-4 treatment[53]. We also revealed reduced *Spp1* (encoding osteopontin) and related signals, which may prevent T-cell exhaustion, as reported recently[54]. Interestingly, CD8+ T cells did not provide substantial benefit to the M002-mediated therapeutic outcome, and CD8+ T cells could not maintain their effector function despite that M002 treatment increased their early effector activity to some extent in our system. Nevertheless, these propositions require further investigation to distinguish if these mechanisms are treatment- or GBM model-dependent.

Lastly, it needs to be pointed out that our study focused on two preclinical GBM models. Given the GBM heterogeneity, future studies will be required to investigate the immune response of oHSV treatment of other murine GBM models. In addition, the exact nature of the immune response elicited by different oHSV variants may differ. Our study explored G207 and M002, which express ICP47 that interacts with the transporter associated with antigen processing-1 and prevents MHCI presentation in human cells but not efficiently in murine cells[55], possibly impacting CD8+ T-cell responses, but also emphasizing CD4+ T-cell responses in our clinical trial patients. In contrast, other oHSV with a deletion of ICP47, such as G47delta, may result in more CD8+ T-cell-mediated responses[42,45]. Likewise, engineered oHSV with γ134.5 expressed under a specific promoter, such as rQNestin34.5 under the nestin promoter[56], may interfere with the MHCII-CD4-dependent response. However, the finding of memory CD4+ T-cell response was corroborated by analysis of our oHSV clinical trial and public GBM patient datasets. Our results revealed the intratumoral CD4+ T-cell differentiation hierarchy during oHSV therapy, which may suggest strategies to develop more effective therapies against GBM.

## Methods

All reagents or resources are listed in Supplemental Data 6, if not indicated in the text.

### Mice

C57BL/6 J (B6), *Tbx21*fl/fl, *Bcl6*fl/fl, *CD4*Cre, *CD4*Cre-ERT2, *Tcrα*−/−, *Rosa26*tdTomato, and MHCII−/− mice (Jackson Labs) were housed in pathogen-free conditions. *GzmB*Cre mice[57], kindly provided by Dr. Dorina Avram at H. Lee Moffitt Cancer Center and Research Institute, were also maintained in pathogen-free conditions at 72 °F ± 2° on a 12 hour day/night cycle lasting from 6am to 6pm with humidity at 52% ± 2%. *Tbx21*fl/fl mice were bred onto *CD4*Cre mice to generate *Tbx21*fl/fl*CD4*Cre mice. *Bcl6*fl/fl mice were bred onto *CD4*Cre-ERT2 mice to generate *Bcl6*fl/fl*CD4*CreER mice. *GzmB*Cre mice were bred onto *Rosa26*tdTomto mice to generate

$GzmB^{Cre}Rosa26^{tdTomto}$ mice. All mice used were 6–11 weeks of age unless otherwise specified. Both sexes (males or females) were randomly included for comparison groups in all experiments in an unblinded fashion. Generally, 3–11 mice were used per group unless otherwise specified in each experiment. All animal experiments were performed in compliance with federal laws and institutional guidelines as approved by the Institutional Animal Care and Use Committee (IACUC) at the University of Alabama at Birmingham (UAB).

## Cell lines and viruses

GFP+GSC005 glioma cells, kindly provided by Dr. Inder M. Verma at the Salk Institute for Biological Studies, were cultured in serum-free Dulbecco's Modified Eagle Medium/F12 Ham (DMEM/F12) containing 2% N2, 1× Penicillin/Streptomycin (100 U/ml and 100 µg/ml), 2 mM GlutaMax, 2.5 µg/ml Heparin, 20 ng/ml EGF, and 20 ng/ml FGF, as described previously[22]. GL261-PVRL1 tumor cells were established by transfecting GL261 cells (obtained from the Division of Cancer Treatment Tumor Repository)[27] with a mouse nectin-1-expressing vector (pCMV-Script-PVRL1) using lipofectamine 3000 (ThermoFisher Scientific) and selected for neomycin resistance with 400 µg/ml G418. To construct the pCMV-Script-PVRL1 vector, the complete coding region sequences were PCR amplified from the mouse PVRL1 cDNA (Origene, MR208266) and were cloned into the pCMV-Script vector (Agilent, 212220) at *BamHI* and *XhoI* sites. The accuracy of all plasmids was confirmed by DNA sequencing. GL261-PVRL1 cells were cultured in DMEM containing 10% FBS, 1× Penicillin/Streptomycin, 1 mM sodium pyruvate, and 200 µg/ml G418. For passaging, Accutase was used as a dissociation reagent. Vero cells (CCL-81), obtained from the American Type Culture Collection, were grown and maintained in MEM containing 7% FBS, as previously reported[16,27]. All cell lines were confirmed pathogen-free via Charles River Research Animal Diagnostic Services, used with three to ten passages, and maintained at 37 °C with 5% $CO_2$. M002, M201, and control R3659 ($\gamma_1$34.5-deleted but does not express IL-12) viruses were constructed and propagated, and the methods for testing the viral entry and replication in tumor cells were adopted, as described previously[16,27,58,59].

## GBM models and oHSV treatment

Mice were anesthetized with an intraperitoneal injection of 100 µl ketamine (100 mg/kg) mixed with xylazine (15 mg/kg) and a subcutaneous injection of 100 µl buprenorphine (0.05–0.1 mg/kg) and 100 µl carprofen (5 mg/kg) before surgery. The heads of the mice were shaved and disinfected with betadine and alcohol swabs. A small incision was then made over the midline of the scalp, followed by drilling a small bur hole in the skull. Each mouse was then placed in a stereotactic head holder (Stoelting). $5 \times 10^4$ GSC005 or $10^5$ GL261-PVRL1 cells were implanted intracranially using a Nanoliter Injector set (Stoelting) in a volume of 3 µl over 3 min via the bur hole over the right frontal cortex 2 mm lateral and 3 mm anterior to the bregma and 3 mm into the striatum of the brain. Following implantation, the bur hole was filled with bone wax, and the incision was closed with Vetbond (3 M Corp.). 11 days (for GSC005) or 10 days (for GL261-PVRL1) after tumor implantation, mice were again anesthetized and given analgesics, as described above. M002 or the control R3659 virus ($10^7$ pfu) or saline (as control) was administered in 2 µl over 2 min through the same bur hole as previously made for tumor implantation. Mice were monitored daily following surgery. In some experiments, mice were intraperitoneally injected with 300 µg of anti-CD4 antibody (GK1.5) or anti-CD8 antibody (2.43) or their isotype control antibody (LTF-2) twice with a 4-day interval to deplete CD4+ T cells or CD8+ T cells, respectively. Some mice were intraperitoneally injected with 100 µl of 1 mg tamoxifen emulsified in sunflower oil once every 24 h for 5 consecutive days. Mice were monitored daily after injection. Tissue samples were harvested on pre-determined experimental endpoints post-implantation of tumor

cells and oHSV treatment. Otherwise, for survival analysis, mice that developed signs of neurologic symptoms as evidenced by general appearance (e.g., unkempt appearance, hunching), loss of avoidance behavior when touched or picked up, failure to eat or drink, or with a loss of >20% of initial body weight, or with <2 body condition score, according to the UAB IACUC guidelines, were euthanized by $CO_2$ inhalation followed by cervical dislocation, making clinical endpoints.

## Cell isolation

The spleen was extracted, and a single-cell suspension was obtained by mashing the spleen between frosted microscope slides. Red blood cells were then removed using the Ammonium-Chloride-Potassium (ACK) lysis buffer and the cell suspension was filtered through a 70 µm filter membrane to eliminate debris. To isolate single cells from brain tumors, tumors were mechanically disassociated into small pieces (<2 mm) followed by digestion for 1 h at 37 °C on a shaker in a dissociation solution (PBS supplemented with 2% FBS, 1 mg/ml Collagenase/Dispase and 0.5 mg/ml DNase I). After passing through a 70 µm cell strainer and thoroughly washing with DMEM/2% FBS, individual digested samples were suspended in a 30% Percoll and centrifuged at $360 \times g$ at 4 °C for 30 min with no brakes. Brain tumor cells and immune cells were collected, washed, and removed from red blood cells using ACK lysis buffer followed by further analysis.

## Flow cytometry and sorting

Single-cell suspension was first stained with the fixable viability dye at 1:1000 in PBS for 20 min. After washing with flow-activated cell sorting (FACS) buffer (PBS/2%FBS), cells were incubated with Fc block at 1:200 for 10 min, followed by staining with staining antibody mixtures for 30 min before washing and flow cytometry analysis. For intracellular staining, cells were fixed and permeabilized using the Foxp3 Staining Buffer Set according to the manufacturer's protocol. Cells were then incubated with intracellular antibodies for 30 min before washing and flow cytometry analysis. All of the steps were performed at 4 °C. To perform tetramer staining, cells were incubated with each tetramer (I-Ab HSV-1 gD 290-302, IPPNWHIPSIQDA; I-Ab mouse Trp-1 113-126, CRPGWRGAACNQKI; I-Ab human CLIP 87-101, PVSKMRMATPLLMQA; all were obtained from NIH tetramer Core Facility) at 1:200 for 3 h at 37 °C, and then stained with fixable viability dye, surface antibodies and finally intracellular antibodies. For intracellular cytokine analysis, cells were stimulated with the BD Leukocyte Activating Cocktail, with BD GolgiPlug for 5 h at 37 °C with 5% $CO_2$, prior to staining, as described above. Cells were acquired on a BD LSR II or FACSymphony using FACSDiva software (BD Biosciences) and analyzed using FlowJo software (Treestar). For cell sorting, single-cell suspensions isolated from brain tumors were enriched for CD45+ cells using the CD45 microbeads. Enriched CD45+ cells were then stained with viability dye and surface antibodies as described above, followed by sorting on a FACSAria II using FACSDiva software. All gating strategies are presented in Supplementary Fig. 11.

## Adoptive transfer

Cells were isolated from brain tumors, enriched for CD45+ cells, and sorted for CD4+CD3+ T cells, as described above. $5 \times 10^4$ CD4+ T cells were then intravenously injected into $Tcr\alpha^{-/-}$ mice that are deficient in T cells, followed by intracranially injected with GSC005 tumor cells 3 days later, as described above. In some experiments, $5 \times 10^4$ CD4+ T cells were transferred into $Tcr\alpha^{-/-}$ mice that had established brain tumors for 25 days. Mice were monitored daily, as described above.

## Specific killing assay with splenocytes

Bulk splenocytes from naive B6 mice were harvested and either pulsed with the Trp-1 peptide at 5 µg/ml or given a mock pulse followed by incubation for 30 min at 37 °C with 5% $CO_2$. After removing excess

peptide by washing with complete DMEM (cDMEM) containing 10% FBS, 1× Penicillin/Streptomycin, 100 mM HEPES, 1 mM sodium pyruvate, 0.1 mM non-essential amino acid and 55 nM 2-Mercaptoethanol, peptide-pulsed cells were stained with 0.1 μM of CFSE and the mock pulsed cells were stained with 1 μM of CFSE, respectively, and were then mixed at a 1:1 ratio. CD4$^+$ T cells were isolated from mice bearing GSC005 tumor that were treated with saline or M002 for 14 days, and then enriched using CD4 microbeads. The mixture of CFSE-labeled cells was co-cultured with CD4$^+$ T cells for 12 hr at 37 °C with 5% $CO_2$ followed by analysis on a flow cytometer. Specific killing of tumor cells was calculated by % specific lysis = 100−((CFSE low/CFSE high) in the presence of effector cells / mean of the three wells containing target cells alone in the absence of effector cells) × 100, as reported previously[60].

### In vitro killing of tumor cells by CD4$^+$ T cells

CD4$^+$ or CD8$^+$ T cells were isolated from mice bearing GSC005 tumor that were treated with saline or M002 for 14 days, and then enriched using CD4 or CD8 microbeads. In some cases, CD4$^+$ T cells were isolated and enriched at day 35 post-treatment. CD4$^+$ or CD8$^+$ T cells were co-cultured with 5000 in vitro-cultured GSC005 tumor cells in an increasing ratio of T cell to tumor cells in cDMEM for 16 hr. The frequency of live CD45$^-$ tumor cells was compared to that of dead CD45$^-$ tumor cells. In some groups, 20 μg/ml anti-MHCII antibody or its isotype control antibody was added to tumor cells 30 min prior to co-culture. Anti-IFNγ, anti-FasL, anti-GzmB, or their respective isotype control antibody (all at 10 μg/ml) was added to CD4$^+$ T cells 30 min (except 1 hr for IFNγ neutralization) prior to co-culture. The specific killing was calculated as ratios by dividing dead over live GSC005 cell numbers in the presence of effector cells.

### In vitro anti-IL6rα blockade

The single-cell suspension of splenocytes from naive B6 mice was obtained, as described above, and was given a 5000 cGy dose of radiation to serve as feeder cells. Following wash with cDMEM, irradiated feeder cells were plated at $3 \times 10^5$ cells in a 96-well round bottom plate, and co-cultured with $10^5$ CD4$^+$ T cells isolated and enriched from GSC005 brain tumors at day 12 post-treatment, as described above. 10 μg/ml Trp-1 peptide and 5 μg/ml anti-IL6rα or its isotype control antibody were added into the co-culture for 80 hr before flow cytometry analysis.

### G207 RNA sequencing dataset analysis

We re-analyzed the public RNA-seq dataset (GSE162643) for our G207 phase I b trial[15] from the GEO repository. Briefly, differentially expressed genes (DEGs) were obtained using the GREIN interactive web pipeline[61] for post-G207 treatment compared to pre-treatment groups using survival days as a covariate. Fold changes of MHCII genes were inferred using manual curation from the DEG list. The Gene Set Enrichment Analysis (GSEA) analysis was done using GSEA software version 4.3.2, which uses predefined gene sets from the Molecular Signatures Database (MSigDB v5.0). For the current analysis, we used KEGG pathways, and a list of ranked genes was calculated as −log10 of $P$ value multiplied by the fold change. Criteria for the selection of genes were limited to the upregulated genes with greater than twofold change and false discovery rate (FDR) significance of <0.05. Pathways were chosen with nominal $P$ < 0.05. For analysis of leading-edge genes from inferred KEGG pathways, genes were selected based on the number of hits in different pathways with a cutoff of >4 pathways from the GSEA software and plotted using the heatmap package in R. The CIBERSORTx algorithm was used to deconvolve bulk gene expression data into cell type proportions using the latest LM22 reference signature matrix downloaded from the CIBERSORTx website (https://cibersortx.stanford.edu/)[62]. Preprocessed gene expression data were input into the CIBERSORTx web interface, and cell fractions were imputed using the default

settings. The analysis was performed separately for each patient sample. Correlations between indicated CD4$^+$ T-cell fractions and clinical outcome (survival days post-G207 treatment) were assessed using Pearson correlation coefficient with $P$ < 0.05.

### Single-cell RNA sequencing and pre-processing

Cells were isolated and pooled from brain tumors of mice ($n = 3$) at day 45 post-treatment with saline or M002, as described above, followed by sorting of CD45$^-$ and CD45$^+$ cells, which were then mixed at 1:1 ratio in each treatment group before single-cell RNA-seq (scRNA-seq) analysis. Briefly, libraries were prepared using (3' library preparation kit v3, 10× Genomics) according to the manufacturer's instructions. Sequencing was performed on Illumina NovaSeq6000 (Illumina Inc.) with an average sequencing depth of over 18,000 reads per cell. Reads from 10× scRNA expression libraries were aligned to mouse genome assembly GRCm38 (mm10) using Cell Ranger Cloud Platform (Cell Ranger Count v7.1.0, 10× Genomics). Quality control metrics, including the number of reads per cell, mitochondrial gene content, and gene detection rate, were calculated to assess the data quality as described elsewhere[63]. Low-quality cells with low gene detection count or high mitochondrial gene content were removed. Additionally, genes detected in fewer than [$n = 10$] cells were excluded. Suspected doublet cells were removed using the DoubletFinder package (version 2.0.4)[64]. Following quality control steps, a total of 15,241 cells were available for downstream analysis.

### Single-cell RNA sequencing data analysis

The above-preprocessed data was normalized and scaled using the Seurat package (v5) functions[65]. Briefly, gene expression counts were transformed to log space and scaled to ensure that the distribution of gene expression was similar across cells. Principal component analysis (PCA) was performed on the scaled data using the RunPCA function. Either the entire preprocessed dataset or CD4 T cells (based on *Ptprc* (encoding CD45) >1 and *Cd4* expression levels >0.5) were utilized for downstream processing depending on the specific analysis. The top principal components (PCs) were selected for downstream analysis of CD4 T cells (PC 15) or the total population (PC 18) based on the elbow plot function. Using the clustree package[66], the Louvain resolution parameter was set to 0.5 or 0.2 for CD4 or total cells, respectively, to control the granularity of clusters. The dimensionality reduction method of the Uniform Manifold Approximation and Projection (UMAP) was applied to visualize the clusters using the RunUMAP function. DEGs for each cluster were identified using the FindMarkers function, comparing each cluster against all other clusters. Different cell populations, including malignant and immune cells, were annotated and visualized for marker genes[24,25] using the clustered dotplot function from the ggSCvis package (https://github.com/junjunlab/ggSCvis). Stacked barplot and volcano plot were generated using the dittoseq package[67], and the EnhancedVolcano package (https://github.com/kevinblighe/EnhancedVolcano), respectively. Unless otherwise stated, statistical significance was defined using a significance threshold of adjusted $P$ < 0.05 (*FDR*-corrected). All data analyses were conducted using R Studio (version 2023.09.0 + 463, R version 4.3.3).

### Gene set enrichment analysis

Single sample geneset enrichment (ssGSEA) and custom GSEA for marker genes were performed using clusterProfiler[68], fgsea[69], and escape[70] packages. Memory T-cell signature (MemS) was adapted as described in Choi et al.[38] and gmt files for GSEA were created using GSEA software[71,72]. Final results, including cell clustering, marker gene expression, and GSEA, were visualized using various plotting functions in Seurat, scCustomize (https://doi.org/10.5281/zenodo.5706430) and GSEAVis (https://github.com/junjunlab/GseaVis) packages. Statistical tests for differential gene expression and enrichment analyses were performed using a non-parametric Wilcoxon rank-sum test built within Seurat, escape, or fgsea packages.

## Velocity mapping

To analyze the expression dynamics in *Foxp3⁻* CD4⁺ T-cell subset, we utilized scVelo (v0.2.5, python package) enabling estimation of RNA velocity of single cells based on transcriptional dynamics of splicing kinetics using a likelihood-based dynamical model[73]. We generated individual loom files for each sample using Cell Ranger and merged saline, and M002 loom files together. Then we fed the merged loom files and UMAP coordinates of single cells generated by Seurat to scVelo to project the RNA velocity vectors onto low-dimension embeddings.

## Pseudotime trajectory inference with Slingshot and tradeseq

To explore the developmental trajectory of CD4⁺ T cells, Slingshot[74] was applied to the clusters of CD4⁺ T cells that were identified using Seurat. Slingshot was initialized on the cluster centroids obtained from the Louvain clustering, using the CD4⁺ T-cell cluster 3 as the starting point as informed by unsupervised velocity mapping. We constructed a minimum spanning tree (MST) connecting cluster centroids in a manner that reflects the underlying data structure, facilitating a logical progression from the starting cluster. From the MST, we inferred potential lineages by identifying paths that represent plausible developmental trajectories. Cells were assigned pseudotime values based on their positions along these inferred lineages, representing their relative progression through a developmental pathway. Following trajectory inference, differential gene expression analysis along the pseudotime trajectories was conducted using the tradeSeq package[75]. The association between gene expression and pseudotime was modeled using negative binomial generalized additive models, with adjustments for cellular library size. This analysis helped identify genes that were differentially expressed along the developmental pathways of CD4⁺ T cells across different treatment conditions. The resulting trajectories were visualized in two-dimensional space using UMAP, or overlaying the inferred pseudotime trajectories along the developmental continuum of condition-specific lineage.

## Cell-cell communication analysis using CellChat

We employed the CellChat package[76] to analyze the cell-cell communication networks among CD4⁺ T cells, tumor cells, microglia, and myeloid cells. The analysis was performed by the identification of signaling pathways and ligand-receptor pairs using the database provided by CellChat (v2.0). Signaling molecules expressed by tumor cells, microglia, and myeloid cells and their respective receptors on total CD4⁺ T cells or cluster 2 subset were identified. The interaction strengths and the significance of these signaling pathways between the different cell types, and between the saline and M002-treated groups were assessed and compared. Communication networks were visualized using the network plotting features of CellChat, which highlighted the predominant signaling pathways altered by the treatment. Statistical significance of the observed differences in communication patterns was determined using CellChat's integrated statistical tests, with p-values adjusted for multiple comparisons using the Benjamini-Hochberg procedure, considering an adjusted *P* value of <0.05 as significant.

## NicheNet ligand-receptor-target analysis

Using the NicheNet package[77], we identified ligands expressed by microglia, myeloid, and tumor cells and their potential receptors on CD4⁺ T cells. The analysis utilized NicheNet's comprehensive database of ligand-receptor interactions, focusing on ligands differentially expressed in response to treatment and their impact on CD4⁺ TCR signaling. NicheNet's gene regulatory network analysis was employed to predict the regulatory effects of identified ligands on CD4⁺ T cells. This approach integrated the ligand activity from microglia, myeloid, and tumor cells with CD4⁺ T-cell gene expression profiles to infer downstream target genes. Predictions included linking active ligands in the TME to receptor-mediated signaling pathways and subsequent gene expression alterations in CD4⁺ T cells. Differential expression analysis

was conducted to compare the gene expression in CD4⁺ T cells between saline-treated and M002-treated groups. This analysis identified genes that were significantly regulated through ligand-receptor interactions differentially activated by the treatments (Top 50 prioritized ligands). Results were visualized using heat maps and network diagrams. Statistical significance was assessed using the Benjamini-Hochberg method to control the *FDR*, with an adjusted *P* value of <0.05 as significant.

## TCGA-GBM survival analysis

Batch-corrected, upper quartile-normalized RNA-seq by expectation maximization data for relevant genes from the TCGA-GBM dataset were obtained using the cSurvival portal (https://tau.cmmt.ubc.ca/cSurvival). Survival time was chosen based on the overall survival (OS) time and OS status. The OS status denotes survival time in days, indicating whether the patient's death was observed (status = 1) or that survival time was censored (status = 0). To investigate the correlation of gene expression with patient survival, samples were grouped into high versus low cohorts based on the exhaustive search model for optimal cutoff as described elsewhere[78]. The Kaplan–Meier method was used to compute survival curves for patients in each group. The log-rank test was used to determine whether patients with different expression values for indicated genes had significantly different survival times ($P < 0.05$).

## Statistics

Statistical analyses were performed using the unpaired Student's *t* test, one-way ANOVA, or log-rank test with GraphPad Prism V9.5 software (Graphpad Inc.), in addition to those specified. Error bars indicate mean ± standard error mean. A *P* value of <0.05 was considered to be statistically significant (*$P < 0.05$, **$P < 0.01$, ***$P < 0.001$, ****$P < 0.0001$). No exclusion of data points was used.

## Reporting summary

Further information on research design is available in the Nature Portfolio Reporting Summary linked to this article.

## Data availability

The G207 RNA-seq data are extracted from the NCBI GEO under accession number GSE162643. The single-cell RNA-seq data are deposited in the NCBI GEO under accession number GSE246895. All data generated or analyzed during this study are included in this article and its supplementary information. Source data are provided with this paper.

## Code availability

All analyses were performed using freely available R packages. Plots and graphs were generated with software and pipelines that have been used in previous papers. No custom codes were used.

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

## Acknowledgements

We thank Dr. Dorina Avram at H. Lee Moffitt Cancer Center and Research Institute for providing *GzmB*^Cre mice, Dr. Inder M. Verma at the Salk Institute for Biological Studies for providing GSC005 cell lines, NIH tetramer Core Facility for providing I-Ab tetramers, and Vidya Sagar Hanumanthu, Dr. Shanrun Liu, the UAB Comprehensive Flow Cytometry and Single-Cell Core, and the UAB Genomics Core for their assistance with FACS analysis, cell sorting, and single-cell RNA-seq analysis. This study was supported by DoD W81XWH-18-1-0315, and the UAB faculty start-up funds (J.W.L.). J.W.L is also supported by NIH R01AI148711, R01AI148711-03S1/04S1 (via NIA), R01CA276190, R21CA278853, DoD HT9425-23-1-0792, American Cancer Society RSG-23-1038722-01-IBCD and Breast Cancer Research Foundation of Alabama.

## Author contributions

J.W.L., J.M.G., and S.G. designed and performed experiments, analyzed data and interpreted the results. S.M. assisted with the single-cell RNA-seq analysis, J.C.C. and M.M.M. assisted with in vivo experiments, L.X.L. generated and characterized GL261-PVRL1 cell lines, S.D.A. propagated viruses. S.G., S.M., and J.M.M. participated in the discussion and manuscript writing. J.W.L. and J.M.G. wrote the paper. J.W.L. conceived and supervised the study.

## Competing interests

J.M.M. received payments from the structured buyout of Catherex, Inc., which was purchased by Amgen in 2015 and no longer exists. J.M.M. also has equity in and has received royalties from Aettis, Inc., which holds frozen oncolytic viral stocks. J.M.M. is a co-owner of Treovir, Inc which holds a small business innovation research fund to execute a clinical trial of G207 in pediatric patients. J.M.M. has served as a consultant for Imugene. J.M.M. holds intellectual property for another oncolytic virus, C134, which has been licensed to Mustang Bio, Inc., but is blinded to the specifics of the relationship. The remaining authors declare no competing interests.
