## [Transparent Peer Review file · Nature Communications]

Oncolytic reprogramming of tumor microenvironment shapes CD4 T-cell memory via the IL6ra-Bcl6 axis for targeted control of glioblastoma

Corresponding Author: Dr Jianmei Leavenworth

Version 0:

Reviewer comments:

Reviewer #1

(Remarks to the Author)

This study on oncolytic herpes simplex virus (HSV) treatment for GBM identified CD4+ T-cells as critical for durable tumor control. OHSV upregulated MHCII on tumor cells, helping CD4+ T-cell tumor elimination and memory response. A Bcl-6 to T-bet ratio and IL6r alpha-Bcl-6 axis in CD4+ T-cells were crucial for this effect. Overall, this is a significant study. We have been trying to uncover why and how CD4+ T cells contribute to treatment efficacy and memory response in immunologically cold GBM models. This study unravels these two critical questions we have been trying to understand. I want to thank the author for finding solutions to those questions, which will help the scientific community working on immunologically cold glioblastoma and oncolytic immunovirotherapy. However, one primary concern regarding this study, specifically the 005 GSC-derived GBM model, should be addressed. M002 is an IL-12 expressing oHSV, whereas their control virus, G207, does not express a cytokine. I am surprised by the fact that they excluded the G207 control virus in all of their mouse studies, and without this proper control, it is impossible to understand the role IL-12 plays here, i.e., how IL-12 expression within the 005 tumors is interrelated with CD4+ T cell-mediated tumor protection and memory response.

Reviewer #2

(Remarks to the Author)

In their manuscript entitled "Oncolytic reprogramming of tumor microenvironment shapes CD4 T-cell memory via the IL6ra-Bcl6 axis for targeted control of glioblastoma", Grimes et al. aim at characterizing the functional consequences of the treatment with oncolytic herpes simplex virus (oHSV) expressing IL-12, using syngeneic mouse models of glioblastoma (GBM). The authors first revisited the previously generated transcriptome datasets from the phase Ib oHSV-G207 clinical trial cohort. They confirmed that genes related to antigen presentation to T-cells are upregulated post-viral inoculation. Consistently, the authors observed increased levels of MHC-II expression in diverse cell types (e.g., tumor cells, myeloid cells) in the 005 syngeneic GBM model when treated with oHSV-expressing IL12 (M002) that can extend animal survival. Through these observations, the authors then concentrated their effort on examining the roles of CD4 T cells as anti-tumor effector cells. They performed adoptive T cell transfer experiments coupled with functional assays and immune profiling and found that T follicular helper (Tfh) and T-helper 1 (Th1) fate trajectories are dynamically involved in the anti-tumor immune response augmented by M002 viral treatment. The authors exploited single-cell RNA-sequencing (scRNA-seq) data to infer cell-cell interactions correlated with the formation of memory T cells. They also employed lineage trajectory analysis using RNA velocity and characterized the relationship between IL6 signaling and Bcl-6-dependent CD4+ T-cells. Overall, this study provides important mechanistic insights into immune responses augmented by HSV-based oncolytic viruses. However, there are several scientific concerns and a significant amount of space for refinement.

Major comments:

1. One key conclusion drawn from this study is that MHC-II-mediated interactions cause CD4 response. However, this was functionally validated in Figure 3b, only for in vitro tumor cell killing but not for T cell activation/ memory formation in vivo caused by the oncolytic virus system. While Figure 5 is likely to support this notion, CellChat analysis is an inference based on the expression of factors that "may" mediate cell-to-cell communications. As such, the data provided reveals associations but not formal proof for the roles of MHC-II expression in shaping CD4-T cell memory formation.
2. It took quite a long time to understand the experimental setups. It is strongly recommended to revise the manuscript with

consistency for better readability and understanding of the complex nature of experiments. For example, Figure 1g diagram says GSC005 was initially injected on Day-11, M002/saline on Day 0,..., and secondary challenge on Day 45. In the survival curve figure, if correctly understood, the x-axis label of Days Post Tumor Implantation is based on the secondary challenge (starting on Day 45). In addition, the text says a secondary challenge was performed on Day 56; "CD4+ T-cells contributed to the protection against tumor regrowth by re-challenging longer survivors from the M002-treated cohort at day 56 after 1st tumor implantation with GSC005 tumor cells in the contralateral hemisphere,..." The figures should be self-explanatory and therefore, audience should be able to comprehend the data.

3. Similarly, in Figure 1g, it is hard to interpret the experiment as what "age-matched mice" means is unclear. The description within the text is written even in a more complicated manner and requires revision. Also, it is unclear whether the animals that received the secondary challenge succumbed to the death due to a tumor arising in the ipsilateral hemisphere or the contralateral one.

4. Once again, Figure 3e and Figure 3g-l is confusing. Figure 3e is used with Tcr α ^{-/-} while others with control. Moreover, the figure legends seem written hastily, and collection time points are not explained properly and should be revised either in the main figure or figure legend (see Figure Legend 3f).

5. Figure 5 and Figure 6, in particular, RNA velocity analysis, seems to be a cherry-pick. What is the description of Cluster 3 and Cluster 4? Is the Cluster 3 in a naïve or activated state? Why is this trajectory analysis valid only for FoxP3-CD4+ T cells? Accordingly, Figures 5f and 6c are redundant. While the Figure 6c legend describes, "Arrows indicate the direction of lineage differentiation from the less differentiated to the more differentiated states.", I was wondering how the author defined the less differentiated versus differentiated state.

Minor comments:

1. Figure 1b: The explanation of the data is missing as to how many samples were used to generate the data and the variance is missing. The use of terminology such as non-T CD45+, other CD45-, and T cells should be explained. The compartments are not clear are there any overlapping cells here?

2. Page 8 - line 213: while MHCII blockade did not have a discernable impact on saline-treated CD4+ T-cells (Fig. 3d) must be Fig. 3b

3. In the main text, page 9, lines 242 and 243, it is unclear what "macrophages and subset" means.

4. Figure 6k: it is difficult to understand what exactly the pink and blue colored illustration for Ratio (Bcl-6: T-bet) describes.

Reviewer #3

(Remarks to the Author)

This is a very interesting manuscript that details the roles of CD4+ve T cells in response to virotherapy. The identification of how M002 encoded IL-12 gene expression is guiding the CD4T cell activation from Treg to TH cells and the use of knock out mouse models to evaluate the effect is done beautifully. Other strengths include the use of transcriptome data from patient tumors treated with oncolytic virus. The study is rigorous, and the detailed analysis of development of CD4T cells after M002 treatment is done beautifully and will likely impact the field.

Some suggestions are below:

CD4 T cell analysis is done in gl261PVR, but its importance by CD4T cell depletion is done in 005 cells? Does the gl261 model also show reduced efficacy with CD4T cell depletion??

The GSC005 model responds to CD4 T cell depletion: was there an increase in CD4 t cells this model with M002 therapy?

Fig 2: quantification of PD1 or lag 3 +ve CD4T cells should also be shown. While PD1 goes down, it appears a lag3+ve population emerges temporally after M002 treatment.

Fig 3B: the symbols for M002 and M002 + anti MHCII are not labelled to be clearly different. I assume the rescued line is with the antibody, but the legend does not discriminate between the dotted and solid line.

Figure 3J: The increased expansion of CD4 T cells in M002 treated mice: how does this compare with control virus treated animals: is this an effect of IL-12 expression by the virus??

Same for Fig 4, which is also missing a control oncolytic virus backbone, to make the statement that the increase is due to M002 encoded IL-12. Either the correct control should be shown, or the statement should be modified.

RNA velocity maps and cell chat analysis were used to show specific CD4 T cell subset changes. The use of knock out mice also bolsters the authors inference on the role of Tfh and TH1 subsets. However the cell chat analysis shows minor to moderate changes in the interaction and strengths of the signals chatting with T cells. This should be discussed.

The authors did not try to modulate IL6 signaling in this pathway, but do talk about the role of IL-sRa in CD4T cells. They discuss recent reports on IL-6 role in GBM, they should also discuss the role of IL-6 in regrowth of residual tumor after virotherapy (PMID: 37114074), where inhibition of AKT signaling led to reduced IL-6/STAT3 and synergized with radiation in immune competent mice.

Reviewer #4

(Remarks to the Author)

Oncolytic HSV has been used to treat some tumors with success. In this study, the authors used preclinical models of glioblastoma combined with oncolytic HSV treatment, which delayed tumor growth and extended survival of GB-bearing mice, and concluded that this improved outcome is mediated by CD4+ T cell recognition of tumor and direct killing. This conclusion is based on upregulation of genes associated with MHC-II-dependent antigen presentation, tumor killing in vitro, and prolonged survival of tumor-bearing lymphopenic mice by adoptive transfer of CD4+ TIL. As a mechanism, based on their finding that M002 (oHSV) treatment increased expression of IL6ra in CD4+ T cells, which inhibits Bcl6 expression, they suggested that the reciprocal regulation between Il6ra and Bcl6 facilitates long-lasting anti-tumor effects through enhanced

memory formation.

Since immunological changes following oHSV challenges in tumors are understudied, the current study provides additional insights into changes in tumors following the treatment, such as a conversion of cold tumors to warm tumors and detailed analysis of phenotypic changes of activated CD4+ T cells. However, there are several major flaws in this study, including little assessment of CD8+ T cell-mediated immunity, no distinction between antiviral versus anti-tumor immunity and effects of CD4+ T cell immunity to tumor growth, which may not be limited to cytotoxicity. It seems reasonable to conclude that MHC-II-dependent recognition of tumor antigen occurs, there is no direct evidence supporting the authors' conclusion that this mainly led to direct killing even though co-cultured peptide-pulsed splenocytes could be killed by CD4+ T cells under unphysiological conditions in vitro. In addition, it is obvious that host CD4+ T cells respond to viral antigen before the clearance, particularly in the case with IL-12 expressing virus, probably besides to tumor antigen. This was barely assessed in all experiments.

Major points:

1. The role of CD8+ T cells: the authors conclusion ignored the contribution of CD8+ T cells for tumor control. While adoptive transfer to lymphopenic mice resulted in delayed tumor growth, this is a very artificial condition. The authors should use CD8+ T cell depletion to determine whether there is a substantial contribution of CD8+ T cells in this model. Furthermore, this reviewer would like the author to test in detail models of GB with an epitope recognized by T cells, which allows them to specifically assess direct anti-tumor immunity by CD4+ and CD8+ T cells, respectively.
2. While CD4+ T cells are capable of killing target cells under certain conditions, the data only showed that CD4+ TIL upregulates GZMB only transiently, not consistent with their continuous killing activity. Since tumors are never rejected in their model, neither slower tumor growth nor prolonged survival are necessarily dependent on killing. These phenotypes can result from cytokine mediated inhibition of tumor growth or changes in stromas. This is also consistent with their observation that there are substantial fraction of surviving cells expressing MHC class II, which should be preferentially killed if CD4+ T cell-mediated killing is the major mechanism. The authors should demonstrate that CD4+ T cell-mediated killing is the major mechanisms of prolonged survival rather than killing-independent effects by transplanting the mixture of MHC-II-sufficient and MHC-II-KO tumors. Alternatively, the authors could set up a similar set of experiment using CD4+ T cell-specific deletion of Gzmb to further show that CD4+ T cell-mediated killing is a major mechanism.
3. In addition, they should assess the time course of CD4+ T cell response to HSV. Obviously, the method they used to detect cytokine production does not distinguish anti-tumor versus antiviral response.
4. This study is heavily dependent on scRNA data in Fig 1a, which was taken 45 days post treatment, namely 56 days after tumor transplantation. At this point majority of control treated mice already died and it is conceivable that there would be substantial changes in tumor sizes between the cohorts. This raises a concern on how meaningful these data are to seek for changes elicited by oHSV. Most of the changes would be indirect.

Minor points:

1. Please clearly describe the pre-gating for all flow data, such as Fig.2. For example, in Fig. 2b, are presented data showing CD4 and Foxp3 expression in pregated CD4+ CD3+ T cells? Otherwise, why would all Foxp3+ cells are CD4(lo/-)?
2. Determine proportions of CD4+ T cells, in which Bcl6 was successfully deleted by CD4-creER to distinguish whether loss of Bcl6 or loss of Bcl6-expressing cells causes the phenotypes in Fig.4
3. Due to the increase in MHC-II in CD4+ T cells, please determine frequency and number of cDC2.
4. Showing Perforin protein expression by CD4+ T cells may support their conclusion that CD4+ TIL acquire killing function which contribute to tumor control.

Reviewer #5

(Remarks to the Author)

Version 1:

Reviewer comments:

Reviewer #1

(Remarks to the Author)

The authors have taken my comments into consideration and provided appropriate responses. No additional comments from my end.

Reviewer #2

(Remarks to the Author)

The authors have revised and significantly improved the manuscript by addressing our concerns.

Reviewer #3

(Remarks to the Author)

The authors have answered all concerns previously brought up by the reviewers.

Reviewer #4

(Remarks to the Author)

In this revised manuscript, the authors have conducted additional experiments to address concerns raised by this reviewer and others. Overall, the authors new data improved the clarity of their finding. However, these data also highlighted that anti-tumor CD4+ T cells are quite distinct between early and late phases in that high Granzyme and perforin expression, which the authors suggested in the initial submission mediate cytotoxicity by CD4+ T cells, is restricted to early phases. The authors response also suggested that the initial response could be directed to virus whereas epitope spreading potentially is the key component of anti-tumor immunity, which unlikely directed towards canonical cytotoxicity. Although the authors responses are overall satisfactory, the data do not support their statements in the abstract and other parts of the manuscript, such as "directly eliminate tumor cells" (line 260, "direct rejection of the tumor" (line 31) and so on.

The reviewer request that the authors clearly and accurately reflect the findings supported by their data and uncertainty which they have not demonstrated in the revised manuscript in addition to the rationale to use single cell data analysis which was conducted in the late time points. Additional experiments are not necessary.

Reviewer #5

(Remarks to the Author)

We thank all the five referees for your careful reviews. All of the comments were thoughtful, well-considered and helpful. Based on all of your suggestions, we performed additional studies to address all the comments and have revised our manuscript accordingly. We believe this has strengthened the manuscript. We have included a point-by-point response to each reviewer as presented below.

Please note: To address the questions raised by the reviewers, we present data newly generated during this revision and designate these figures as “new Fig XX”. We also refer to figures displayed in the current revised manuscript, including those original figures with reordered numbers, as “Fig XX” or “Supplementary Fig XX”.

Reviewer #1 (Remarks to the Author): with expertise in oncolytic therapy, glioblastoma, cancer immunology

This study on oncolytic herpes simplex virus (HSV) treatment for GBM identified CD4+ T-cells as critical for durable tumor control. OHSV upregulated MHCII on tumor cells, helping CD4+ T-cell tumor elimination and memory response. A Bcl-6 to T-bet ratio and IL6r alpha-Bcl-6 axis in CD4+ T-cells were crucial for this effect. Overall, this is a significant study. We have been trying to uncover why and how CD4+ T cells contribute to treatment efficacy and memory response in immunologically cold GBM models. This study unravels these two critical questions we have been trying to understand. I want to thank the author for finding solutions to those questions, which will help the scientific community working on immunologically cold glioblastoma and oncolytic immunovirotherapy. However, one primary concern regarding this study, specifically the 005 GSC-derived GBM model, should be addressed. M002 is an IL-12 expressing oHSV, whereas their control virus, G207, does not express a cytokine. I am surprised by the fact that they excluded the G207 control virus in all of their mouse studies, and without this

proper control, it is impossible to understand the role IL-12 plays here, i.e., how IL-12 expression within the 005 tumors is interrelated with CD4+ T cell-mediated tumor protection and memory response.

Response: The point regarding the role of M002-derived IL-12 in the regulation of CD4+ T cell-mediated tumor control and memory response is well taken. To address this question, we compared the CD4+ T-cell response in

mice bearing GSC005 tumors that were treated with saline or M002 or R3659, the M002 noncytokine parent virus^{1, 2} at different time points post-treatment (days 4.5, 14 and 35). R3659 treatment

abundance of CD4+ T-cells and decreased the relative abundance of Treg cells within the tumor microenvironment (TME) compared to saline-treated controls, albeit with a lesser extent than M002 treatment (new Fig. 1a, b). Importantly, the expression levels of granzyme B (GzmB), perforin and T-bet were only significantly increased in CD4+ T-cells from mice treated with M002, but not R3659, despite that both R3659 and M002 treatments had significantly more CD4+ T-cells expressing IFNγ and TNFα compared to saline controls at day 4.5 (new Figs. 1c-e and 2a). Although both R3659 and M002 treatments increased Bcl-6 expression at day 4.5, the ratio of Bcl-6 to T-bet was not changed in R3659-treated cells compared to saline controls but a significant reduction was noted in M002-treated cells (new Fig. 2b, c), consistent with our previous findings (Fig. 4c). These results suggest that M002 treatment with the additional IL-12 had the better capacity to enhance CD4+ T-cell effector activity compared to its parent virus.

When CD4⁺ T-cell response at the later time point (day 35 post-treatment) was analyzed, we found that the Tcf-1 expression and the proportion of CD4⁺ T-cells expressing Bcl-6 and IL6ra were only significantly increased, while the expression of Tim3 was also only significantly reduced in mice treated with M002, but not in mice treated with R3659 (new Fig. 3a-c). Notably, when CD4⁺ T-cells isolated from brain tumor at day 28 post-treatment were incubated with irradiated splenocytes pulsed with Trp-1 peptide, M002-treated CD4⁺ T-cells expressed much higher levels of GzmB (new Fig. 3d). These results suggest that M002 treatment with the additional IL-12 also had the better capacity to improve CD4⁺ T-cell memory response compared to its parent virus.

The increased effector and memory responses by CD4⁺ T-cells in M002-treated mice compared to its noncytokine parent virus has provided explanations for why M002 inhibits

glioma growth and improves mice survival more effectively than R3659, as we reported previously². All of results are provided and discussed (lines 183-196, 328-332, 450-452, 500-502).

Reviewer #2 (Remarks to the Author): with expertise in glioblastoma, tumor microenvironment

In their manuscript entitled “Oncolytic reprogramming of tumor microenvironment shapes CD4 T-cell memory via the IL6ra-Bcl6 axis for targeted control of glioblastoma”, Grimes et al. aim at characterizing the functional consequences of the treatment with oncolytic herpes simplex virus (oHSV) expressing IL-12, using syngeneic mouse models of glioblastoma (GBM). The authors first revisited the previously generated transcriptome datasets from the phase Ib oHSV-G207 clinical trial cohort. They confirmed that genes related to antigen presentation to T-cells are upregulated post-viral inoculation. Consistently, the authors observed increased levels of MHC-II expression in diverse cell types (e.g., tumor cells, myeloid cells) in the 005 syngeneic GBM model when treated with oHSV-expressing IL12 (M002) that can extend animal survival. Through these observations, the authors then concentrated their effort on examining the roles of CD4 T cells as anti-tumor effector cells. They performed adoptive T cell transfer experiments coupled with functional assays and immune profiling and found that T follicular helper (Tfh) and T-helper 1 (Th1) fate trajectories are dynamically involved in the anti-tumor immune response augmented by M002 viral treatment. The authors exploited single-cell RNA-sequencing (scRNA-seq) data to infer cell-cell interactions correlated with the formation of memory T cells. They also

employed lineage trajectory analysis using RNA velocity and characterized the relationship between IL6 signaling and Bcl-6-dependent CD4⁺ T-cells. Overall, this study provides important mechanistic insights into immune responses augmented by HSV-based oncolytic viruses. However, there are several scientific concerns and a significant amount of space for refinement.

Major comments:

1. One key conclusion drawn from this study is that MHC-II-mediated interactions cause CD4 response. However, this was functionally validated in Figure 3b, only for in vitro tumor cell killing but not for T cell activation/ memory formation in vivo caused by the oncolytic virus system. While Figure 5 is likely to support this notion, CellChat analysis is an inference based on the expression of factors that “may” mediate cell-to-cell communications. As such, the data provided reveals associations but not formal proof for the roles of MHC-II expression in shaping CD4-T cell memory formation.

Response: To provide additional evidence supporting the role of MHCII expression in shaping CD4⁺ T-cell

response in our GSC005 and oHSV system, we administered the anti-MHCII antibody into mice bearing GSC005 tumor with the treatment of saline or M002 to block the interactions between MHCII and CD4⁺ T-cells in vivo (new Fig. 4a), based on the strategy reported previously³. Although MHCII blockade did not substantially change the overall abundance of CD4⁺ T-cells, there was a significant reduction of CD4⁺ T-cells expressing IFN γ and TNF α in mice treated with M002 and anti-MHCII compared to mice treated with M002 and its isotype control antibody (new Fig. 4b, c). MHCII blockade also

reduced the Tcf-1 expression and the proportion of CD4⁺ T-cells expressing Bcl-6 and IL6ra in mice treated with M002 (new Fig. 4d, e). Interestingly, the above results were not obvious in saline-treated mice comparing those treated with anti-MHCII antibody to mice treated with its isotype control antibody. These findings support our in vitro findings (Fig. 3b) and CellChat analysis (Fig. 5g-k and Supplementary Fig. 8f-j), and provide additional support that the MHCII-mediated interaction is important for CD4⁺ T-cell effector and memory response after M002 treatment in our experimental system. All of results are provided and discussed (lines 417-428).

2. It took quite a long time to understand the experimental setups. It is strongly recommended to revise the manuscript with consistency for better readability and understanding of the complex nature of experiments. For example, Figure 1g diagram says GSC005 was initially injected on Day-11, M002/saline on Day 0,..., and secondary challenge on Day 45. In the survival curve figure, if correctly understood, the x-axis label of Days Post Tumor Implantation is based on the secondary challenge (starting on Day 45). In addition, the text says a secondary challenge was performed on Day 56; "CD4⁺ T-cells contributed to the protection against tumor regrowth by re-challenging longer survivors from the M002-treated cohort at day 56 after 1st tumor implantation with GSC005 tumor cells in the contralateral hemisphere,..". The figures should be self-explanatory and therefore, audience should be able to comprehend the data.

Response: We have revised the manuscript text, figure, figure legends throughout the manuscript for clarity and consistency, including revisions for Fig. 1g (lines 134-143).

3. Similarly, in Figure 1g, it is hard to interpret the experiment as what "age-matched mice" means is unclear. The description within the text is written even in a more complicated manner and requires revision. Also, it is unclear whether the animals that received the secondary challenge succumbed to the death due to a tumor arising in the ipsilateral hemisphere or the contralateral one.

Response: The manuscript has been revised to provide clarifications (line 134-143). Briefly, M002 treated mice (6-7 week old) that survived from the 1st tumor implantation at day 45 were given the secondary tumor challenge, and therefore these mice were at ~ 3 months old. Naïve mice of similar age (~3 months) were implanted with GSC005 at the same day when M002-treated mice were re-challenged. Anti-CD4 or its isotype control antibodies were given one day prior and 3 days after the re-challenge (for M002-treated mice) or after the GSC005 implantation (for naïve mice). We cannot determine which tumor led to the death of the animal or which tumor was the main cause of the animal reaching its euthanasia conditions, as required by our IACUC (see Methods, lines 613-617).

4. Once again, Figure 3e and Figure 3g-l is confusing. Figure 3e is used with Tcr α ^{-/-} while others with control. Moreover, the figure legends seem written hastily, and collection time points are not explained properly and should be revised either in the main figure or figure legend (see Figure Legend 3f).

Response: The figure and figure legends have been revised to provide clarifications and consistency for Fig. 3e, Fig. 3f and Fig. 3g-k.

5. Figure 5 and Figure 6, in particular, RNA velocity analysis, seems to be a cherry-pick. What is the description of Cluster 3 and Cluster 4? Is the Cluster 3 in a naïve or activated state? Why is this trajectory analysis valid only for FoxP3-CD4+ T cells? Accordingly, Figures 5f and 6c are redundant. While the Figure 6c legend describes, “Arrows indicate the direction of lineage differentiation from the less differentiated to the more differentiated states.”, I was wondering how the author defined the less differentiated versus differentiated state.

Response: Cluster 4 is the 2nd Treg cluster according to its relatively high expression of *Foxp3* (Supplementary Fig. 8c). Cluster 3 is an effector-like cluster based on its relatively high expression of *Tbx21* and *Cxcr6*, but relatively low levels of *Pdcd1* and *Havcr2* compared to cluster 1 (Supplementary Fig. 8c). All of this description has been added into the text (lines 386-389).

Our initial trajectory analysis included all clusters, including both Treg clusters 1 and 4. This analysis showed an additional differentiation path to these Treg clusters. To focus on our analysis of effector/memory response of CD4⁺ T-cells, we excluded Treg clusters in our current trajectory analysis.

However, this exclusion did not change our results and conclusion, as the differentiation path to Treg clusters 1 and 4 was distinct from the differentiation paths from cluster 3 to other clusters representative of Foxp3-CD4⁺ T-cell subsets (see figure on the right).

Figures 5f and 6c are not redundant. The violin plot in Figure 5f showed the relative expression of genes related to memory and exhaustion within each cluster comparing M002-treated CD4⁺ T-cells to their saline control cells. Figure 6c revealed these genes across pseudo time reflecting CD4⁺ T-cell differentiation. The differentiation path is determined by the RNA splicing and un-splicing status within each individual cell, as indicated by arrows based on the RNA velocity algorithm⁴. We revised this to “Arrows indicate the direction of lineage differentiation for individual cells”.

Minor comments:

1. Figure 1b: The explanation of the data is missing as to how many samples were used to generate the data and the variance is missing. The use of terminology such as non-T CD45+, other CD45-, and T cells should be explained. The compartments are not clear are there any overlapping cells here?

Response: The sample numbers along with additional information related to scRNA-seq analysis has been added to the Methods section (lines 706-708). A total of 15,241 cells from both saline- (9,934 cells) and M002-treated (5,307 cells) groups were analyzed. The explanation for the terminology used for each cell compartment was provided in the legend of Fig. 1a. The naming strategy was justified according to the representative gene signatures of each cell compartment, based on the publications^{5, 6}, as shown in Supplementary Fig. 1b-d.

2. Page 8 - line 213: while MHCII blockade did not have a discernable impact on saline-treated CD4+ T-cells (Fig. 3d) must be Fig. 3b

Response: Thank you. It is now corrected.

3. In the main text, page 9, lines 242 and 243, it is unclear what “macrophages and subset” means.

Response: Additional text (lines 262-263) was added to clarify this.

4. Figure 6k: it is difficult to understand what exactly the pink and blue colored illustration for Ratio (Bcl-6: T-bet) describes.

Response: Additional text was added to the Figure 6k legend to clarify that blue color indicates the cold, *i.e.* an immunologically cold tumor, and pink color indicates hot, *i.e.* an immunologically hot tumor induced by oHSV treatment.

Reviewer #3 (Remarks to the Author): with expertise in oncolytic therapy, glioblastoma, cancer immunology

This is a very interesting manuscript that details the roles of CD4+ve T cells in response to virotherapy. The identification of how M002 encoded IL-12 gene expression is guiding the CD4T cell activation from Treg to TH cells and the use of knock out mouse models to evaluate the effect is done beautifully. Other strengths include the use of transcriptome data from patient tumors treated with oncolytic virus. The study is rigorous, and the detailed analysis of development of CD4T cells after M002 treatment is done beautifully and will likely impact the field.

Some suggestions are below:

CD4 T cell analysis is done in gl261PVR, but its importance by CD4T cell depletion is done in 005 cells? Does the gl261 model also show reduced efficacy with CD4T cell depletion??

Response: We performed the depletion of CD4⁺ T-cells in mice implanted with GL261-PVRL1 and treated with saline or M002. We observed a slight but significant increase in survival in M002-treated mice compared to saline-treated mice, which received the anti-CD4 isotype control antibody. This increase in survival was ablated in mice given the anti-CD4 depletion antibody (new Fig. 5). These results suggest that CD4⁺ T-cells also contributed to the M002-mediated therapeutic benefit in the GL261-PVRL1 model. Results are added to lines 123-127).

The GSC005 model responds to CD4 T cell depletion: was there an increase in CD4 t cells this model with M002 therapy?

Response: Yes. Figure. 2a has shown that M002 treatment significantly increased CD4⁺ T-cells compared to saline control.

Fig 2: quantification of PD1 or lag 3 +ve CD4T cells should also be shown. While PD1 goes down, it appears a lag3+ve population emerges temporally after M002 treatment.

Response: As suggested by the reviewer, we have quantified the frequency of PD-1⁺CD4⁺ or Lag-3⁺CD4⁺ T-cells (see figure on the right). The portion of CD4⁺ T-cells expressing PD-1 was relatively stable in both saline and M002-treated groups, while the portion of CD4⁺ T-cells expressing Lag-3 was increased over the time in

both groups. However, the portion of CD4⁺ T-cells expressing PD-1 or Lag-3 were lower in M002-treated group than saline-treated controls. Importantly, the frequency of PD-1⁺Lag-3⁺ double-positive CD4⁺ T-cells, which reflect cells tending to be more exhausted due to the expression of multiple inhibitory receptors⁷, were reduced in M002-treated group (Fig. 2e).

Fig 3B: the symbols for M002 and M002 + anti MHCII are not labelled to be clearly different. I assume the rescued line is with the antibody, but the legend does not discriminate between the dotted and solid line.

Response: We have provided clarifications in the legend of Figure 3b.

Figure 3J: The increased expansion of CD4 T cells in M002 treated mice: how does this compare with control virus treated animals: is this an effect of IL-12 expression by the virus??

Response: We have performed a new set of experiments to address the role of M002-derived IL-12 in the regulation of CD4⁺ T cell-mediated tumor control and memory response. Please refer to response to reviewer 1 (pages 1-3, new **Figs. 1-3**).

Same for Fig 4, which is also missing a control oncolytic virus backbone, to make the statement that the increase is due to M002 encoded IL-12. Either the correct control should be shown, or the statement should be modified.

Response: Please refer to our response above.

RNA velocity maps and cell chat analysis were used to show specific CD4 T cell subset changes. The use of knock out mice also bolsters the authors inference on the role of Tfh and TH1 subsets. However the cell chat analysis shows minor to moderate changes in the interaction and strengths of the signals chatting with T cells. This should be discussed.

Response: We acknowledge that the number of interactions and the strength of interaction, as judged by the numbers, appear moderate (Fig. 5g and Supplementary Fig. 8f). However, the major finding from our CellChat analysis is that the interaction of CD4 with MHCII on tumor cells was only present in M002-treated CD4⁺ T-cells, but was absent in saline-treated CD4⁺ T-cells (Fig. 5i, j and Supplementary Fig. 8h, i). All of these have been described or discussed in lines 408-410 and 492-497.

The authors did not try to modulate IL6 signaling in this pathway, but do talk about the role of IL-sRa in CD4T cells. They discuss recent reports on IL-6 role in GBM, they should also discuss the role of IL-6 in regrowth of residual tumor after virotherapy (PMID: 37114074), where inhibition of AKT signaling led to reduced IL-6/STAT3 and synergized with radiation in immune competent mice.

Response: Thank you. We have included the reference PMID: 37114074 and discussed its relevance to our findings (lines 517-518).

Reviewer #4 (Remarks to the Author): with expertise in CD4 T cell biology

Oncolytic HSV has been used to treat some tumors with success. In this study, the authors used preclinical models of glioblastoma combined with oncolytic HSV treatment, which delayed tumor growth and extended survival of GB-bearing mice, and concluded that this improved outcome is mediated by CD4⁺ T cell recognition of tumor and direct killing. This conclusion is based on upregulation of genes associated with MHC-II-dependent antigen presentation, tumor killing in vitro, and prolonged survival of tumor-bearing lymphopenic mice by

adoptive transfer of CD4⁺ TIL. As a mechanism, based on their finding that M002 (oHSV) treatment increased expression of IL6ra in CD4⁺ T cells, which inhibits Bcl6 expression, they suggested that the reciprocal regulation between Il6ra and Bcl6 facilitates long-lasting anti-tumor effects through enhanced memory formation.

Since immunological changes following oHSV challenges in tumors are understudied, the current study provides additional insights into changes in tumors following the treatment, such as a conversion of cold tumors to warm tumors and detailed analysis of phenotypic changes of activated CD4⁺ T cells. However, there are several major flaws in this study, including little assessment of CD8⁺ T cell-mediated immunity, no distinction between antiviral versus anti-tumor immunity and effects of CD4⁺ T cell immunity to tumor growth, which may not be limited to cytotoxicity. It seems reasonable to conclude that MHC-II-dependent recognition of tumor antigen occurs, there is no direct evidence supporting the authors' conclusion that this mainly led to direct killing even though co-cultured peptide-pulsed splenocytes could be killed by CD4⁺ T cells under unphysiological conditions in vitro. In addition, it is obvious that host CD4⁺ T cells respond to viral antigen before the clearance, particularly in the case with IL-12 expressing virus, probably besides to tumor antigen. This was barely assessed in all experiments.

Major points:

1. The role of CD8⁺ T cells: the authors conclusion ignored the contribution of CD8⁺ T cells for tumor control. While adoptive transfer to lymphopenic mice resulted in delayed tumor growth, this is a very artificial condition. The authors should use CD8⁺ T cell depletion to determine whether there is a substantial contribution of CD8⁺ T cells in this model. Furthermore, this reviewer would like the author to test in detail models of GB with an epitope recognized by T cells, which allows them to specifically assess direct anti-tumor immunity by CD4⁺ and CD8⁺ T cells, respectively.

Response: Our analysis of CD8⁺ T-cells revealed the non-significant increase of intratumoral CD8⁺ T-cells in M002-treated mice compared to saline-treated mice across the time course (Supplementary Fig. 3b). To further address if CD8⁺ T-cells contributed to tumor control in our system, we performed CD8⁺ T-cell depletion, as suggested by the reviewer. The CD8⁺ T-cell depletion using the anti-CD8 depleting antibody 2.43 had no significant impact on the survival of saline-treated mice (new Fig. 6). We also did not observe a significant effect of CD8⁺ T-cell depletion on the survival of M002-treated mice, although M002

New Figure 6. Depletion of CD8⁺ T-cells did not affect the therapeutic benefit of M002 treatment. Mice (n = 8 per group) were implanted with 50,000 GSC005 tumor cells at day -11, and were given either M002 (blue) or saline (black) at day 0. Mice were also received either anti-CD8 (ΔCD8, dotted lines) or an isotype control (Iso Ctrl, solid lines) one day before and three days after treatment. Kaplan-Meier analysis of median survival as M002 Iso Ctrl, 93 days; Saline Iso ctrl, 53 days; M002 ΔCD8, undefined days and Saline ΔCD8, 54 days, as the status on the day of resubmission.

treatment consistently prolonged mice survival compared to the saline-treated groups (new Fig. 6). Additionally, we further analyzed the effector activity of CD8⁺ T-cells. Both R3659, the M002 noncytokine parent virus (see response to Reviewer 1, page 1), and M002 treatments increased the proportion of CD8⁺ T-cells expressing IFN γ and TNF α , albeit with no significance compared to the saline-treated group (new Fig. 7a). M002 but not

R3659 treatment also significantly increased the expression of GzmB, but not perforin, in CD8⁺ T-cells at day 4.5 (new Fig. 7b, c). However, Lag-3 expression was not reduced and Tcf-1 expression was not significantly increased in CD8⁺ T-cells at day 35 post-M002 treatment (new Fig. 7d, e). Compared to

CD4⁺ T-cell response post-M002 treatment (new Figs. 1,3), these results suggest that M002 treatment increased

the effector activity of CD8⁺ T-cells at the early time point, but was likely not able to sustain their effector function at the late stage. In fact, M002-treated CD8⁺ T-cells appeared to have reduced killing activity than saline-treated cells at higher effector to target ratios, as shown in Fig. 3d. Thus, CD8⁺ T-cells provided the limited benefit to M002 treatment in our experimental system (lines 128-131, 191-196, 450-452, 545-547).

As requested by the reviewer, we have generated GSC005-OVA cell lines that expressed the ovalbumin (OVA) in GSC005 cells (new Fig. 8a), which contained epitopes

that can be recognized by OT-I CD8⁺ T-cells and OT-II CD4⁺ T-cells. To specifically assess their anti-tumor activity in our experimental system, we adoptively transferred a same number of purified OT-I CD8⁺ T-cells or OT-II CD4⁺ T-cells into B6 mice implanted with GSC005-OVA cells and treated with saline or M002 (new Fig. 8b). M002 treatment increased OT-II CD4⁺ T-cells expressing IFN γ and TNF α , but reduced OT-II CD4⁺ T-cells expressing CD44 and KLRG1 compared to saline treatment (new Fig. 8c, d). Interestingly, these changes were not obvious in transferred OT-I CD8⁺ T-cells, and more OT-I CD8⁺ T-cells expressed KLRG1 than OT-II CD4⁺ T-cells under conditions of both saline and M002 treatments (new Fig. 8c, d). Given that KLRG1 expression marks intratumoral CD4⁺ and CD8⁺ T-cells associated with tumor progression and often restricts anti-tumor immunity^{8, 9}, these results suggest that transferred CD4⁺ T-cells had the better capacity to maintain effector activity than transferred CD8⁺ T-cells in response to M002 treatment when they both were specific to the same tumor antigen, although additional detailed studies are needed to further support this finding.

2. While CD4⁺ T cells are capable of killing target cells under certain conditions, the data only showed that CD4⁺ TIL upregulates GZMB only transiently, not consistent with their continuous killing activity. Since tumors are never rejected in their model, neither slower tumor growth nor prolonged survival are necessarily dependent on killing. These phenotypes can result from cytokine mediated inhibition of tumor growth or changes in stromas. This is also consistent with their observation that there are substantial fraction of surviving cells expressing MHC class II, which should be preferentially killed if CD4⁺ T cell-mediated killing is the major mechanism. The authors should demonstrate that CD4⁺ T cell-mediated killing is the major mechanisms of prolonged survival rather than killing-independent effects by transplanting the mixture of MHC-II-sufficient and MHC-II-KO tumors. Alternatively, the authors could set up a similar set of experiment using CD4⁺ T cell-specific deletion of Gzmb to further show that CD4⁺ T cell-mediated killing is a major mechanism.

Response: Thank you for raising this point. Our data do not fully exclude the contribution of cytokines, specifically IFN γ , to the anti-tumor activity of CD4⁺ T-cells after M002 treatment, as IFN γ produced by CD4⁺ T-cells increased tumor MHCII and could potentiate their anti-tumor activity (Supplementary Fig. 5c-e, lines 531-533). M002-treated CD4⁺ T-cells displayed enhanced polyfunctional anti-tumor activity and their tumor killing capacity required the interaction of tumor MHCII and CD4⁺ T-cells based on our in vitro assays (Fig. 3b-d). We agree with this reviewer that we need to experimentally demonstrate the importance of tumor MHCII in mediating the effector activity of CD4⁺ T-cells in vivo. We respectfully disagree with the experimental approach by transplanting the mixed MHCII-WT and MHCII-KO tumors, as our experimental system entails an oHSV-mediated initial killing of tumor cells followed by immune-mediated anti-tumor responses, and the possible bystander killing of MHCII-KO tumors will complicate our interpretation. To this end, we designed an alternative

strategy. We purified CD4⁺ T-cells isolated from brain tumors of saline- or M002-treated mice and transferred them at a same number into MHCII KO (MHCII^{-/-}) mice that were implanted with GSC005 tumor cells at 10 days earlier (new Fig. 9a). In this strategy, MHCII was only present on tumor cells. The anti-MHCII antibody was also injected into one group of mice given M002-treated CD4⁺ T-cells to block the interaction between tumor

MHCII and CD4⁺ T-cells, based on the strategy reported previously³. We observed fewer surviving CD45-GFP⁺ tumor cells in mice transferred with M002-treated CD4⁺ T-cells than mice given saline-treated CD4⁺ T-cells (new Fig. 9b). The frequency of surviving tumor cells was increased in mice injected with the anti-MHCII antibody, which was not significant compared to mice given saline-treated CD4⁺ T-cells (new Fig. 9b). These results suggest that the *in vivo* anti-tumor activity of CD4⁺ T-cells required MHCII expression on tumor cells (lines 299-309).

Thank the reviewer for the suggestion regarding the use of a CD4⁺ T-cell-specific deletion of GzmB. Unfortunately, GzmB^{fl} mice are not commercially available, and breeding these mice requires an additionally significant amount of time. Although the significant increase of GzmB was only observed early after M002 treatment, after rechallenge with irradiated Trp-1-pulsed splenocytes, GzmB expression was increased (Fig. 6h and new Fig. 3d), and these CD4⁺ T-cells were able to mediate anti-tumor response in vitro (Fig. 3b-d) or in vivo (Fig. 3e-k, and new Fig. 9b). Our in vitro GzmB blockade assay also indicates that CD4⁺ T-cell-mediated anti-tumor response may not substantially require GzmB (Supplementary Fig. 5e). These results suggest that the effector activity of CD4⁺ T-cells may be subject to M002 treatment and the tumor microenvironmental cues in our experimental system, and it is likely that CD4⁺ T-cell effector activity is differentiation stage-dependent, and other anti-tumor mechanisms need to be investigated to better understand their function (lines 536-541).

3. In addition, they should assess the **time course** of CD4⁺ T cell response to HSV. Obviously, the method they used to detect cytokine production does not distinguish anti-tumor versus antiviral response.

Response: The point regarding the anti-tumor versus antiviral response in our system is well taken. However, distinguishing anti-tumor versus anti-viral response in the field of oncolytic virotherapy against GBM is challenging, as epitopes of GBM tumor cells and oncolytic virus (OV) are largely unclear, although it is commonly accepted that OV-mediated killing of tumor cells induces subsequent epitope spreading, which can facilitate the T-cell-mediated anti-tumor response^{10, 11}. We tried

to address CD4⁺ T-cell antigen-specific responses in our experimental system by using Trp1 as a representative tumor-associated antigen, and using glycoprotein D (gD) as a representative HSV antigen and their respective tetramers. As requested by the reviewer, we analyzed the HSVgD-specific CD4⁺ T-cell response over the time of course. The frequency of HSVgD-tetramer-positive CD4⁺ T-cells was increased at day 14 compared to day 4.5 and day 35 post-M002 treatment, although no significance was achieved (new Fig. 10). We acknowledge that this analysis with one HSV epitope is not sufficient to provide a clear conclusion, and additional approaches, including single cell TCR repertoire analysis and antigen epitope analysis of GBM tumor cells, are required to further dissect the nature of CD4⁺ T-cell responses in our experimental system.

4. This study is heavily dependent on scRNA data in Fig 1a, which was taken 45 days post treatment, namely 56 days after tumor transplantation. At this point majority of control treated mice already died and it is conceivable that there would be substantial changes in tumor sizes between the cohorts. This raises a concern on how meaningful these data are to seek for changes elicited by oHSV. Most of the changes would be indirect.

Response: The changes of tumor and the tumor microenvironment at day 45 post-M002 treatment are results of both direct oHSV-mediated tumor lysis and subsequent anti-tumor immune responses, as our experimental system entails that OV-mediated killing of tumor cells induces subsequent epitope spreading, which facilitates the anti-tumor immune responses¹⁰⁻¹². The analysis of immune changes at a later time point (e.g., day 45) post-M002 treatment has allowed us to understand what immune components contributed to the sustained beneficial outcome of this therapy.

Minor points:

1. Please clearly describe the pre-gating for all flow data, such as Fig.2. For example, in Fig. 2b, are presented data showing CD4 and Foxp3 expression in pre-gated CD4⁺ CD3⁺ T cells? Otherwise, why would all Foxp3⁺ cells are CD4(10/-)?

Response: Yes, Fig. 2b showing CD4 and Foxp3 is pre-gated on CD4⁺CD3⁺ T-cells, described as CD45^{hi}CD4⁺CD3⁺Foxp3⁺ Treg cells in the legend. We provided the general gating strategy in Supplementary Fig. 11, and also described the sub-gating for all flow data by listing the makers used in the text or respective legends.

2. Determine proportions of CD4⁺ T cells, in which Bcl6 was successfully deleted by CD4-creER to distinguish whether loss of Bcl6 or loss of Bcl6-expressing cells causes the phenotypes in Fig.4

Response: The frequency of total CD4⁺CD3⁺ T-cells in *Bcl6^{fl/fl}CD4^{CreER}* mice was not significantly different from that in *Bcl6^{fl/fl}* mice either after saline or M002 treatment (Supplementary Fig. 7h). However, Bcl6 MFI was reduced in *Bcl6^{fl/fl}CD4^{CreER}* mice (Supplementary Fig. 7f). We also quantified Bcl6⁺CD4⁺ T-cells, as suggested by the reviewer. As shown in Supplementary Fig. 7g (new Fig. 11), the frequency of Bcl6⁺CD4⁺ T-cells were also reduced in *Bcl6^{fl/fl}CD4^{CreER}* mice, suggesting that both loss of Bcl6 and Bcl6-expressing CD4⁺ T-cells may contribute to the survival difference in Fig. 4.

3. Due to the increase in MHC-II in CD4⁺ T cells, please determine frequency and number of cDC2.

Response: We quantified the frequency and numbers of live CD45^{hi}CD11c⁺MHCII⁺CD11b⁺ cells, which were not significantly different between M002-treated and saline-treated mice (new Fig. 12).

4. Showing Perforin protein expression by CD4+ T cells may support their conclusion that CD4+ TIL acquire killing function which contribute to tumor control.

Response: We analyzed intracellular perforin expression in CD4+ T-cells. The frequency of perforin+ CD4+ T-cells was significantly increased at day 4.5 in mice treated with M002 compared to those treated with saline control (new Fig. 1e).

Reviewer #5 (Remarks to the Author):

Thank you!

References

- Markert JM, Cody JJ, Parker JN, Coleman JM, Price KH, Kern ER, Quenelle DC, Lakeman AD, Schoeb TR, Palmer CA, Cartner SC, Gillespie GY, Whitley RJ. Preclinical Evaluation of a Genetically Engineered Herpes Simplex Virus Expressing Interleukin-12. *Journal of Virology*. 2012;86(9):5304-13. doi: 10.1128/JVI.06998-11.
- Hellums EK, Markert JM, Parker JN, He B, Perbal B, Roizman B, Whitley RJ, Langford CP, Bharara S, Gillespie GY. Increased efficacy of an interleukin-12-secreting herpes simplex virus in a syngeneic intracranial murine glioma model. *Neuro Oncol*. 2005;7(3):213-24. Epub 2005/08/02. doi: 10.1215/S1152851705000074. PubMed PMID: 16053696; PMCID: PMC1871915.
- Quezada SA, Simpson TR, Peggs KS, Merghoub T, Vider J, Fan X, Blasberg R, Yagita H, Muranski P, Antony PA, Restifo NP, Allison JP. Tumor-reactive CD4(+) T cells develop cytotoxic activity and eradicate large established melanoma after transfer into lymphopenic hosts. *J Exp Med*. 2010;207(3):637-50. Epub 20100215. doi: 10.1084/jem.20091918. PubMed PMID: 20156971; PMCID: PMC2839156.
- Bergen V, Lange M, Peidli S, Wolf FA, Theis FJ. Generalizing RNA velocity to transient cell states through dynamical modeling. *Nat Biotechnol*. 2020;38(12):1408-14. Epub 20200803. doi: 10.1038/s41587-020-0591-3. PubMed PMID: 32747759.
- Xiao Y, Wang Z, Zhao M, Deng Y, Yang M, Su G, Yang K, Qian C, Hu X, Liu Y, Geng L, Xiao Y, Zou Y, Tang X, Liu H, Xiao H, Fan R. Single-Cell Transcriptomics Revealed Subtype-Specific Tumor Immune Microenvironments in Human Glioblastomas. *Front Immunol*. 2022;13:914236. Epub 20220520. doi: 10.3389/fimmu.2022.914236. PubMed PMID: 35669791; PMCID: PMC9163377.
- Yeo AT, Rawal S, Delcuze B, Christofides A, Atayde A, Strauss L, Balaj L, Rogers VA, Uhlmann EJ, Varma H, Carter BS, Boussiotis VA, Charest A. Single-cell RNA sequencing reveals evolution of immune landscape during glioblastoma progression. *Nat Immunol*. 2022;23(6):971-84. Epub 20220527. doi: 10.1038/s41590-022-01215-0. PubMed PMID: 35624211; PMCID: PMC9174057.
- Blackburn SD, Shin H, Haining WN, Zou T, Workman CJ, Polley A, Betts MR, Freeman GJ, Vignali DA, Wherry EJ. Coregulation of CD8+ T cell exhaustion by multiple inhibitory receptors during chronic viral infection. *Nat Immunol*. 2009;10(1):29-37. Epub 20081130. doi: 10.1038/ni.1679. PubMed PMID: 19043418; PMCID: PMC2605166.

8. Ager CR, Zhang M, Chaimowitz M, Bansal S, Tagore S, Obradovic A, Jugler C, Rogava M, Melms JC, McCann P, Spina C, Drake CG, Dallos MC, Izar B. KLRG1 marks tumor-infiltrating CD4 T cell subsets associated with tumor progression and immunotherapy response. *J Immunother Cancer*. 2023;11(9). doi: 10.1136/jitc-2023-006782. PubMed PMID: 37657842; PMCID: PMC10476134.
9. Li L, Wan S, Tao K, Wang G, Zhao E. KLRG1 restricts memory T cell antitumor immunity. *Oncotarget*. 2016;7(38):61670-8. doi: 10.18632/oncotarget.11430. PubMed PMID: 27557510; PMCID: PMC5308681.
10. Chen X, Zhao J, Yue S, Li Z, Duan X, Lin Y, Yang Y, He J, Gao L, Pan Z, Yang X, Su X, Huang M, Li X, Zhao Y, Zhang X, Li Z, Hu L, Tang J, Hao Y, Tian Q, Wang Y, Xu L, Huang Q, Cao Y, Chen Y, Zhu B, Li Y, Bai F, Zhang G, Ye L. An oncolytic virus delivering tumor-irrelevant bystander T cell epitopes induces anti-tumor immunity and potentiates cancer immunotherapy. *Nat Cancer*. 2024. Epub 20240412. doi: 10.1038/s43018-024-00760-x. PubMed PMID: 38609488.
11. Kaufman HL, Kohlhapp FJ, Zloza A. Oncolytic viruses: a new class of immunotherapy drugs. *Nat Rev Drug Discov*. 2015;14(9):642-62. Epub 2015/09/02. doi: 10.1038/nrd4663. PubMed PMID: 26323545; PMCID: PMC7097180 authors have no competing interests to disclose.
12. Ma R, Li Z, Chiocca EA, Caligiuri MA, Yu J. The emerging field of oncolytic virus-based cancer immunotherapy. *Trends Cancer*. 2023;9(2):122-39. Epub 2022/11/20. doi: 10.1016/j.trecan.2022.10.003. PubMed PMID: 36402738; PMCID: PMC9877109.

Reviewer #4 (Remarks to the Author):

In this revised manuscript, the authors have conducted additional experiments to address concerns raised by this reviewer and others. Overall, the authors new data improved the clarity of their finding. However, these data also highlighted that anti-tumor CD4+ T cells are quite distinct between early and late phases in that high Granzyme and perforin expression, which the authors suggested in the initial submission mediate cytotoxicity by CD4+ T cells, is restricted to early phases. The authors response also suggested that the initial response could be directed to virus whereas epitope spreading potentially is the key component of anti-tumor immunity, which unlikely directed towards canonical cytotoxicity. Although the authors responses are overall satisfactory, the data do not support their statements in the abstract and other parts of the manuscript, such as “directly eliminate tumor cells” (line 260, “direct rejection of the tumor” (line 31) and so on. The reviewer request that the authors clearly and accurately reflect the findings supported by their data and uncertainty which they have not demonstrated in the revised manuscript in addition to the rationale to use single cell data analysis which was conducted in the late time points. Additional experiments are not necessary.

Response: Thank you for the comments. As suggested, we have revised the manuscript to accurately report and discuss our findings with a particular attention to the CD4+ T-cell-mediated tumor control (revised abstract and lines 73, 196, 216, 233, 466, 525, 531, 536-537). We have also included the rationale for using the single cell RNA sequencing data at the late time point post-oHSV treatment (lines 90-91).